# AUTOMATING CONTINUAL LEARNING

## ABSTRACT

General-purpose learning systems should improve themselves in open-ended fashion in ever-changing environments. Conventional learning algorithms for neural networks, however, suffer from catastrophic forgetting (CF)—previously acquired skills are forgotten when a new task is learned. Instead of hand-crafting new algorithms for avoiding CF, we use our novel Automated Continual Learning (ACL) to train self-referential neural networks to meta-learn their own in-context continual (meta-)learning algorithms. ACL encodes all desiderata—good performance on both old and new tasks—into its learning objectives. We demonstrate the effectiveness and promise of ACL on multiple few-shot and standard image classification tasks adopted for continual learning: Mini-ImageNet, Omniglot, FC100, MNIST-families, and CIFAR-10.[1]

## 1 INTRODUCTION

Enemies of memories are other memories (Eagleman, 2020). Continually-learning artificial neural networks (NNs) are memory systems where their *weights* store memories of task-solving skills or programs, and their *learning algorithm* is responsible for memory read/write operations. Conventional learning algorithms—used to train NNs in the standard scenarios where all training data is available "at once" as opposed to "sequentially" in the case of continual learning (CL)—are known to be inadequate for continual learning. They suffer from the so-called catastrophic forgetting (CF; McCloskey & Cohen (1989); Ratcliff (1990); French (1999)) problem, where the NNs forget, or rather, the learning algorithm erases, previously acquired skills, in exchange of learning to solve a new task. Naturally, a certain degree of forgetting is unavoidable when the memory capacity is limited, and the amount of things to remember exceeds such an upper bound. In general, however, capacity is not the fundamental cause of CF; typically, the same NNs, suffering from CF when trained on two tasks sequentially, can perform well on both tasks when they are jointly trained on the two tasks at once instead (see, e.g., Irie et al. (2022a)). The real root of CF lies in the learning algorithm as a memory mechanism. A "good" CL algorithm should preserve previously acquired knowledge while also leveraging previous learning experiences to improve future learning, by maximally exploiting the limited memory space of model parameters. All of this is the decision-making problem of a learning algorithm. In fact, we can not blame the conventional learning algorithms for causing CF, since they are not aware of such a problem. They are designed to train NNs for a given task at hand; they treat each learning experience independently (they are stationary up to certain momentum parameters in certain optimizers), and ignore any potential influence of current learning on past or future learning experiences. Effectively, more sophisticated algorithms previously proposed against CF (Kortge, 1990; French, 1991), such as elastic weight consolidation (Kirkpatrick et al., 2017; Schwarz et al., 2018) or synaptic intelligence (Zenke et al., 2017), often introduce manually-designed constraints as regularization terms to explicitly penalize current learning for deteriorating knowledge acquired in past learning.

Here, instead of hand-crafting learning algorithms for continual learning, we train self-referential neural networks (Schmidhuber, 1992a; 1987) to meta-learn their own 'in-context' continual learning algorithms. We train them through gradient descent on learning objectives that reflect desiderata for continual learning algorithms—good performance on both old and new tasks, including forward and backward transfer. In fact, by extending the standard settings of few-shot or meta-learning based on sequence-processing NNs (Hochreiter et al., 2001; Younger et al., 1999; Cotter & Conwell, 1991; 1990; Mishra et al., 2018), the continual learning problem can also be formulated as a long-span sequence processing task (Irie et al., 2022c). Corresponding sequences can be obtained by

---

[1]Here we will add a link to our public GitHub code repository upon acceptance

concatenating multiple few-shot/meta-learning sub-sequences, where each sub-sequence consists of input/target examples corresponding to the task to be learned in-context. As we'll see in Sec. 3, this setting also allows us to seamlessly express classic desiderata for continual learning (knowledge preservation, forward/backward transfer, etc.) as part of objective functions of the meta-learner.

Once formulated as such a sequence-learning task, we let gradient descent search for CL algorithms achieving the desired CL behaviors in the program space of NN weights. In principle, all typical challenges of CL—such as the stability-plasticity dilemma (Grossberg, 1982)—are automatically discovered and handled by the gradient-based program search process. Once trained, CL is automated through recursive self-modification dynamics of the trained NN, without requiring any human intervention such as adding extra regularization or even setting hyper-parameters for continual learning. Therefore, we call our method, Automated Continual Learning (ACL).

ACL requires training settings and datasets similar to those of few-shot/meta learning problems: training sequences are constructed by shuffling target labels for various combinations of underlying class categories, such that each such sequence represents a new learning experience for the model. Our experiments focus on supervised image classication, and make use of the standard few-shot image classification datasets for meta-training, namely, Mini-ImageNet (Vinyals et al., 2016; Ravi & Larochelle, 2017), Omniglot (Lake et al., 2015), and FC100 (Oreshkin et al., 2018), while we also meta-test on standard image classification datasets including MNIST families (LeCun et al., 1998; Xiao et al., 2017) and CIFAR-10 (Krizhevsky, 2009). While these datasets remain in the realm of toy tasks, they allow us to demonstrate the effectiveness and promise of our ACL principle.

## 2 BACKGROUND

Here we briefly review some background concepts that are essential for describing our method in Sec. 3: continual learning and its desiderata (Sec. 2.1), few-shot/meta learning via sequence processing (Sec. 2.2), and linear Transformer/fast weight programmer architectures (Sec. 2.3) that are foundations of the self-referential neural network we use in our experiments.

### 2.1 CONTINUAL LEARNING

Continual or lifelong learning is a special form of multi-task learning (Thrun, 1998; Caruana, 1997). In conventional multi-task learning scenarios, all datasets for different tasks are available *at once*, and NNs are trained jointly on all of them (e.g., training batches mix examples from different datasets without any particular order). In contrast, in continual learning settings, presentation of tasks/datasets is *sequential*; an NN training process only has access to one task/dataset at a time. When a dataset for a new task becomes available, we lose access to the training examples from the previous task, and so on.

The main focus of this work is on continual learning in *supervised* learning settings. In addition, we focus on the realm of CL methods that keep model sizes constant (unlike certain CL methods that incrementally add more parameters as more tasks are presented; see, e.g., Rusu et al. (2016)), and do not make use of any external replay memory (used in other CL methods; see, e.g., Robins (1995); Rolnick et al. (2019); Zhang et al. (2022)). High-level principles we discuss here also transfer to reinforcement learning settings (Ring, 1994), but our experiments focus on supervised learning.

Classic desiderata for a CL system (see, e.g., Lopez-Paz & Ranzato (2017); Veniat et al. (2021)) are typically summarized as good performance on three metrics: *classification accuracies* on each dataset (or their average), *backward transfer* (which measures the impact of learning a new task on the model's performance on previous tasks; e.g., catastrophic forgetting is a negative backward transfer), and *forward transfer* (impact of learning a task for the model's performance on a future task). From a broader perspective of meta-learning systems, we may also measure other effects such as *learning acceleration* (i.e., whether the system leverages previous learning experiences to accelerate future learning); here we only briefly discuss this as our primary focus remains the classic CL metrics above.

### 2.2 FEW-SHOT/META-LEARNING VIA SEQUENCE LEARNING

In Sec. 3, we'll formulate continual learning as a long-span sequence processing task. This is a direct extension of the classic few-shot/meta learning formulated as sequence learning problem. In fact, since the seminal works by Hochreiter et al. (2001); Younger et al. (1999); Cotter & Conwell (1991;

1990) (see also Naik & Mammone (1992)), many sequence processing neural networks (see, e.g., Bosc (2015); Santoro et al. (2016); Duan et al. (2016); Wang et al. (2017); Munkhdalai & Yu (2017); Munkhdalai & Trischler (2018); Miconi et al. (2018; 2019); Munkhdalai et al. (2019); Kirsch & Schmidhuber (2021); Sandler et al. (2021); Huisman et al. (2023) including Transformers (Vaswani et al., 2017) as in Mishra et al. (2018)) have been trained as a meta-learner (Schmidhuber, 1987; 1992a) that learn by observing sequences of training examples (i.e., pairs of inputs and their labels) in-context. Here we briefly review such a formulation.

Let $d$, $N$, $K$, $P$ be positive integers. In sequential $N$-way $K$-shot classification settings, a sequence processing NN with a parameter vector $\theta \in \mathbb{R}^P$ observes a pair $(\boldsymbol{x}_t, y_t)$ where $\boldsymbol{x}_t \in \mathbb{R}^d$ is the input and $y_t \in \{1, ..., N\}$ is its label at each step $t \in \{1, ..., N \cdot K\}$, corresponding to $K$ examples for each one of $N$ classes. After the presentation of these $N \cdot K$ examples (often called the *support set*), one extra input $\boldsymbol{x} \in \mathbb{R}^d$ (often called the *query*) is fed to the model without its true label but an "unknown label" token $\varnothing$ (number of input labels accepted by the model is thus $N + 1$). The model is trained to predict its true label, i.e., the parameters of the model $\theta$ are optimized to maximize the probability $p(y|(\boldsymbol{x}_1, y_1), ..., (\boldsymbol{x}_{N \cdot K}, y_{N \cdot K}), (\boldsymbol{x}, \varnothing); \theta)$ of the correct label $y \in \{1, ..., N\}$ of the input query $\boldsymbol{x}$. Since class-to-label associations are randomized and unique to each sequence $((\boldsymbol{x}_1, y_1), ..., (\boldsymbol{x}_{N \cdot K}, y_{N \cdot K}), (\boldsymbol{x}, \varnothing))$, each such a sequence represents a new (few-shot or meta) learning experience to train the model. To be more specific, this is the *synchronous* label setting of Mishra et al. (2018) where the learning phase (observing examples $(\boldsymbol{x}_1, y_1), ..., (\boldsymbol{x}_{N \cdot K}, y_{N \cdot K})$) is separated from the prediction phase (predicting label $y$ given $(\boldsymbol{x}, \varnothing)$). We opt for this variant in our experiments as we empirically find this (at least in our specific settings) more stable than the *delayed* label setting (Hochreiter et al., 2001) where the model has to make a prediction for every input, and the label is fed to the model with a delay of one time step.

## 2.3 SELF-REFERENTIAL WEIGHT MATRICES OR RECURSIVE LINEAR TRANSFORMERS

Our method (Sec. 3) can be applied to any sequence-processing NN architectures in principle. Nevertheless, certain architectures naturally fit better to parameterize a self-improving continual learner. Here we use the *modern self-referential weight matrix* (SRWM; Irie et al. (2022c)) to build a generic self-modifying NN. An SRWM is a weight matrix (WM) that sequentially modifies itself as a response to a stream of input observations (Schmidhuber, 1992a; 1993). The modern SRWM belongs to the family of linear Transformers a.k.a. Fast Weight Programmers (FWPs; Schmidhuber (1991; 1992b); Katharopoulos et al. (2020); Choromanski et al. (2021); Peng et al. (2021); Schlag et al. (2021); Irie et al. (2021)). Linear Transformers and FWPs are an important class of the now popular Transformers (Vaswani et al., 2017): unlike the standard Transformers whose state size linearly grows with the context length, the state size of FWPs is constant w.r.t. sequence length (like in the standard RNNs). This is an important property for in-context continual learning, since, conceptually, we want such a CL system to continue to learn for an arbitrarily long, lifelong time span. Moreover, the duality between linear attention and FWPs (Schlag et al., 2021)—and likewise, between linear attention and linear layers trained by the gradient descent learning algorithm (Irie et al., 2022a; Aizerman et al., 1964)—have played a key role in certain theoretical analyses of in-context learning capabilities of Transformers (von Oswald et al., 2023a; Dai et al., 2023).

The dynamics of an SRWM (Irie et al., 2022c) are described as follows. Let $d_{\text{in}}$, $d_{\text{out}}$, $t$ be positive integers, and $\otimes$ denote outer product. At each time step $t$, an SRWM $\boldsymbol{W}_{t-1} \in \mathbb{R}^{(d_{\text{out}} + 2 * d_{\text{in}} + 1) \times d_{\text{in}}}$ observes an input $\boldsymbol{x}_t \in \mathbb{R}^{d_{\text{in}}}$, and outputs $\boldsymbol{y}_t \in \mathbb{R}^{d_{\text{out}}}$, while also updating itself to $\boldsymbol{W}_t$:

$$[\boldsymbol{y}_t, \boldsymbol{k}_t, \boldsymbol{q}_t, \beta_t] = \boldsymbol{W}_{t-1}\boldsymbol{x}_t \tag{1}$$

$$\boldsymbol{v}_t = \boldsymbol{W}_{t-1}\phi(\boldsymbol{q}_t); \quad \bar{\boldsymbol{v}}_t = \boldsymbol{W}_{t-1}\phi(\boldsymbol{k}_t) \tag{2}$$

$$\boldsymbol{W}_t = \boldsymbol{W}_{t-1} + \sigma(\beta_t)(\boldsymbol{v}_t - \bar{\boldsymbol{v}}_t) \otimes \phi(\boldsymbol{k}_t) \tag{3}$$

where $\boldsymbol{v}_t, \bar{\boldsymbol{v}}_t \in \mathbb{R}^{(d_{\text{out}} + 2 * d_{\text{in}} + 1)}$ are value vectors, $\boldsymbol{q}_t \in \mathbb{R}^{d_{\text{in}}}$ and $\boldsymbol{k}_t \in \mathbb{R}^{d_{\text{in}}}$ are query and key vectors, and $\sigma(\beta_t) \in \mathbb{R}$ is the learning rate. $\sigma$ and $\phi$ denote sigmoid and softmax functions respectively. $\phi$ is typically also applied to $\boldsymbol{x}_t$ in Eq. 1; here we follow Irie et al. (2022c)'s few-shot image classification setting, and use the variant without it. Eq. 3 corresponds to a rank-one update of the SRWM, from $\boldsymbol{W}_{t-1}$ to $\boldsymbol{W}_t$, through the *delta learning rule* (Widrow & Hoff, 1960; Schlag et al., 2021) where the self-generated patterns, $\boldsymbol{v}_t$, $\phi(\boldsymbol{k}_t)$, and $\sigma(\beta_t)$, play the role of *target*, *input*, and *learning rate* of the learning rule respectively. The delta rule in FWPs is typically reported to be crucial broadly across many practical tasks (Schlag et al., 2021; Irie et al., 2021; 2022b; Irie & Schmidhuber, 2023b).

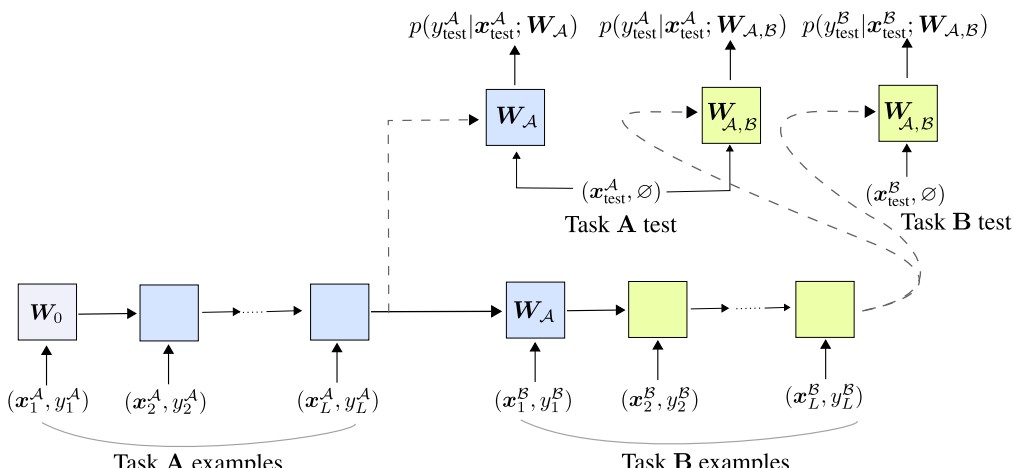

Figure 1: An illustration of meta-training in Automated Continual Learning (ACL) for a self-referential/modifying weight matrix $\boldsymbol{W}_0$. Weights $\boldsymbol{W}_{\mathcal{A}}$ obtained by observing examples for Task A (*blue*) are used to predict a test example for Task A. Weights $\boldsymbol{W}_{\mathcal{A},\mathcal{B}}$ obtained by observing examples for Task A then those for Task B (*yellow*) are used to predict a test example for Task A (backward transfer) as well as a test example for Task B (forward transfer).

The initial weight matrix $\boldsymbol{W}_0$ is the only trainable parameters of this layer, that encodes the initial self-modification algorithm. In practice, we use the layer above as a direct replacement to the self-attention layer in the Transformer architecture (Vaswani et al., 2017); we also use the multi-head version of the SRWM computation above. For further details, we refer to Irie et al. (2022c).

## 3 METHOD

**Task Formulation.** We formulate continual learning as a long-span sequence learning task. Consider two "training" tasks **A** and **B** to be learned sequentially (as we'll see, this can also be straightforwardly extended to training using three or more tasks). We denote the respective training datasets as $\mathcal{A}$ and $\mathcal{B}$, and test sets as $\mathcal{A}'$ and $\mathcal{B}'$. Let $D$, $N$, $K$, $L$ denote positive integers. We assume that each datapoint in these datasets consists of one input feature $\boldsymbol{x} \in \mathbb{R}^D$ of dimension $D$ (here we generically denote $\boldsymbol{x}$ as a vector, but it is an image in all our experiments) and one label $y \in \{1, ..., N\}$ denoting one out of $N$ classes (that is, these are $N$-way classification tasks). We further consider two sequences of $L$ training examples $\left((\boldsymbol{x}_1^{\mathcal{A}}, y_1^{\mathcal{A}}), ..., (\boldsymbol{x}_L^{\mathcal{A}}, y_L^{\mathcal{A}})\right)$ and $\left((\boldsymbol{x}_1^{\mathcal{B}}, y_1^{\mathcal{B}}), ..., (\boldsymbol{x}_L^{\mathcal{B}}, y_L^{\mathcal{B}})\right)$ sampled from the respective training sets $\mathcal{A}$ and $\mathcal{B}$. In practice, $L = NK$ where $K$ is the number of training examples for each class (1 out of $N$). By concatenating these two sequences, we obtain one long sequence representing a continual learning example to be presented to the model as an input sequence. Now we also need to introduce test examples. We assume a single test example (hence, without index) for each task: $(\boldsymbol{x}^{\mathcal{A}'}, y^{\mathcal{A}'})$ and $(\boldsymbol{x}^{\mathcal{B}'}, y^{\mathcal{B}'})$ respectively; let us further simplify the notation and denote them as $(\boldsymbol{x}_{\text{test}}^{\mathcal{A}}, y_{\text{test}}^{\mathcal{A}})$ and $(\boldsymbol{x}_{\text{test}}^{\mathcal{B}}, y_{\text{test}}^{\mathcal{B}})$ instead. In the next section, we describe how these test examples are used to construct the learning objectives to train the model.

Our model is a self-referential neural network that modifies its own weight matrices as a function of input observations. To simplify the notation, we denote the *state* of our self-referential NN as a *single* SRWM $\boldsymbol{W}_*$ (even though, in practice, it may have many of them) where we'll replace $*$ by various symbols representing the context/inputs it has observed. Given a training sequence $\left((\boldsymbol{x}_1^{\mathcal{A}}, y_1^{\mathcal{A}}), ..., (\boldsymbol{x}_L^{\mathcal{A}}, y_L^{\mathcal{A}}), (\boldsymbol{x}_1^{\mathcal{B}}, y_1^{\mathcal{B}}), ..., (\boldsymbol{x}_L^{\mathcal{B}}, y_L^{\mathcal{B}})\right)$, our model consumes one input at a time, from left to right, in the auto-regressive fashion. Let $\boldsymbol{W}_{\mathcal{A}}$ denote the state of the SRWM that has consumed the first part of the sequence corresponding to the examples from Task **A**, i.e., $(\boldsymbol{x}_1^{\mathcal{A}}, y_1^{\mathcal{A}}), ..., (\boldsymbol{x}_L^{\mathcal{A}}, y_L^{\mathcal{A}})$, and let $\boldsymbol{W}_{\mathcal{A},\mathcal{B}}$ denote the state of our SRWM having observed the entire sequence.

**ACL Meta-Training Objectives.** The ACL objective function consists in tasking the model to correctly predict the test examples of all tasks learned so far at each task boundaries. That is, in the case of two-task scenario described above (learning Task **A** then Task **B**), we use the weight matrix

$W_{\mathcal{A}}$ to predict the label $y_{\text{test}}^{\mathcal{A}}$ from input $(\boldsymbol{x}_{\text{test}}^{\mathcal{A}}, \varnothing)$, and we use the weight matrix $W_{\mathcal{A},\mathcal{B}}$ to predict the label $y_{\text{test}}^{\mathcal{B}}$ from input $(\boldsymbol{x}_{\text{test}}^{\mathcal{B}}, \varnothing)$ *as well as* the label $y_{\text{test}}^{\mathcal{A}}$ from input $(\boldsymbol{x}_{\text{test}}^{\mathcal{A}}, \varnothing)$. By letting $p(y|\boldsymbol{x}; W_*)$ denote the model's output probability for label $y \in \{1, .., N\}$ given input $\boldsymbol{x}$ and model weights/state $W_*$, the ACL objective function can be expressed as:

$$\underset{\theta}{\text{minimize}} - \left( \log(p(y_{\text{test}}^{\mathcal{A}}|\boldsymbol{x}_{\text{test}}^{\mathcal{A}}; W_{\mathcal{A}})) + \log(p(y_{\text{test}}^{\mathcal{B}}|\boldsymbol{x}_{\text{test}}^{\mathcal{B}}; W_{\mathcal{A},\mathcal{B}})) + \log(p(y_{\text{test}}^{\mathcal{A}}|\boldsymbol{x}_{\text{test}}^{\mathcal{A}}; W_{\mathcal{A},\mathcal{B}})) \right) \quad (4)$$

for an input training sequence $\left( (\boldsymbol{x}_1^{\mathcal{A}}, y_1^{\mathcal{A}}), ..., (\boldsymbol{x}_L^{\mathcal{A}}, y_L^{\mathcal{A}}), (\boldsymbol{x}_1^{\mathcal{B}}, y_1^{\mathcal{B}}), ..., (\boldsymbol{x}_L^{\mathcal{B}}, y_L^{\mathcal{B}}) \right)$ (which can be easily extended to mini-batches with multiple such sequences), where $\theta$ denotes the model parameters (for the SRWM layer, it is the initial weights $W_0$). Figure 1 illustrates the overall training process of ACL.

The ACL objective function above (Eq. 4) is simple but encapsulates desiderata for continual learning (reviewed in Sec. 2.1). The last term of Eq. 4 with $p(y_{\text{test}}^{\mathcal{A}}|\boldsymbol{x}_{\text{test}}^{\mathcal{A}}; W_{\mathcal{A},\mathcal{B}})$ (or schematically $\boldsymbol{p}(\mathcal{A}'|\mathcal{A},\mathcal{B})$) optimizes for *backward transfer*: (1) remembering the first task **A** after learning **B** (combatting catastrophic forgetting), and (2) leveraging learning of **B** to improve performance on a past task **A**. The second term of Eq. 4, $p(y_{\text{test}}^{\mathcal{B}}|\boldsymbol{x}_{\text{test}}^{\mathcal{B}}; W_{\mathcal{A},\mathcal{B}})$ (or schematically $\boldsymbol{p}(\mathcal{B}'|\mathcal{A},\mathcal{B})$), optimizes *forward transfer* leveraging the past learning experience of **A** to improve predictions in the second task **B**, in addition to simply learning to solve Task **B** from the corresponding training examples. To complete, the first term of Eq. 4 is a simple, single-task meta-learning objective for Task **A**.

**Overall Model Architecture.** As we mention in Sec. 2, in our NN architecture, the core sequential dynamics of continual learning are learned by the self-referential layers. However, as an image-processing NN, our model makes use of a vision backend consisting of multiple convolutional layers. In practice, we use the "Conv-4" architecture of Vinyals et al. (2016) typically used in the context of few-shot learning. Overall, the model takes an image as input, process it through a feedforward convolutional NN, whose output is fed to the SRWM-layer block. Note that this is one of the limitations of this work. While our architecture with fixed vision components still allows us to experimentally demonstrate the principle of ACL, more general ACL should make use of self-modifying NNs that also learn to modify the vision components. One straightforward architecture fitting the bill is an MLP-mixer architecture (Tolstikhin et al. (2021); built of several linear layers), where all linear layers are replaced by the self-referential linear layers of Sec. 2.3. While we implemented such a model, it turned out to be too slow for us to conduct corresponding experiments. We will include our code of self-referential MLP-mixers in our public repository, but for further experiments, we leave the future work with such an architecture using more efficient CUDA kernels.

Another crucial architectural aspect that is specific to continual/multi-task image processing is the choice of normalization layers (see also Bronskill et al. (2020)). Typical convolutional NNs used in few-shot learning (e.g., Vinyals et al. (2016)) contain batch normalization (BN; Ioffe & Szegedy (2015)) layers. In our preliminary experiments, as expected, we found instance normalization (IN; Ulyanov et al. (2016)) to generalize much better than BN layers in our CL setting. Thus, all BN layers in our models are replaced by IN layers.

## 4 EXPERIMENTS

**Common Settings.** All tasks are configured to be a 5-way classification task. Not only this is one of the classic configurations for few-shot learning tasks, but this also allows us to keep the overall computational costs of our experiments reasonable (at the same time, we'll also discuss this as a limitation in Sec. 5). For standard datasets such as MNIST, we split the dataset into sub-datasets of disjoint classes (Srivastava et al., 2013): for example for MNIST which is originally a 10-way classification task, we split it into two 5-way tasks, one consisting of images of class '0' to '4' ('MNIST-04'), and another one made of class '5' to '9' images ('MNIST-59'). When we refer to a dataset without specifying the class range, we refer to the first sub-set. Unless otherwise indicated, we concatenate 15 examples for each class for each task in the context for both meta-training and meta-testing (resulting in sequences of length 75 for each task). All images are resized to $32 \times 32$-size 3-channel images, and normalized according to the original dataset statistics. Appendix A provides further details.

### 4.1 TWO-TASK EVALUATION

Here we show the essence of our method (Sec. 3) in the two-task setting. We consider two meta-training task combinations: Omniglot (Lake et al., 2015) and Mini-ImageNet (Vinyals et al., 2016;

Table 1: 5-way classification accuracies using 15 examples for each class in the context (i.e., as meta-testing training examples). Each row corresponds to a single model. **bold** numbers highlight cases where in-context catastrophic forgetting is avoided through ACL.

| Meta-Training Tasks | | | Meta-Test Tasks: Context/Train (top) & Test (bottom) | | | | | |
|---|---|---|---|---|---|---|---|---|
| | | | A | A → B | | B | B → A | |
| Task A | Task B | ACL | A | B | A | B | A | B |
| Omniglot | Mini-ImageNet | No | $97.6 \pm 0.2$ | $52.8 \pm 0.7$ | $22.9 \pm 0.7$ | $52.1 \pm 0.8$ | $97.8 \pm 0.3$ | $20.4 \pm 0.6$ |
| | | Yes | $98.3 \pm 0.2$ | $54.4 \pm 0.8$ | $\mathbf{98.2} \pm 0.2$ | $54.8 \pm 0.9$ | $98.0 \pm 0.3$ | $\mathbf{54.6} \pm 1.0$ |
| FC100 | Mini-ImageNet | No | $49.7 \pm 0.7$ | $55.0 \pm 1.0$ | $21.3 \pm 0.7$ | $55.1 \pm 0.6$ | $49.9 \pm 0.8$ | $21.7 \pm 0.8$ |
| | | Yes | $53.8 \pm 1.7$ | $52.5 \pm 1.2$ | $\mathbf{46.2} \pm 1.3$ | $59.9 \pm 0.7$ | $45.5 \pm 0.9$ | $\mathbf{53.0} \pm 0.6$ |

Table 2: Similar to Table 1 above but using MNIST and CIFAR-10 (unseen domains) for meta-testing.

| Meta-Training Tasks | | | Meta-Test Tasks: Context/Train (top) & Test (bottom) | | | | | |
|---|---|---|---|---|---|---|---|---|
| | | | MNIST | MNIST → CIFAR-10 | | CIFAR-10 | CIFAR-10 → MNIST | |
| Task A | Task B | ACL | MNIST | CIFAR-10 | MNIST | CIFAR-10 | MNIST | CIFAR-10 |
| Omniglot | Mini-ImageNet | No | $71.1 \pm 4.0$ | $49.4 \pm 2.4$ | $43.7 \pm 2.3$ | $51.5 \pm 1.4$ | $68.9 \pm 4.1$ | $24.9 \pm 3.2$ |
| | | Yes | $75.4 \pm 3.0$ | $50.8 \pm 1.3$ | $\mathbf{81.5} \pm 2.7$ | $51.6 \pm 1.3$ | $77.9 \pm 2.3$ | $\mathbf{51.8} \pm 2.0$ |
| FC100 | Mini-ImageNet | No | $60.1 \pm 2.0$ | $56.1 \pm 2.3$ | $17.2 \pm 3.5$ | $54.4 \pm 1.7$ | $58.6 \pm 1.6$ | $21.2 \pm 3.1$ |
| | | Yes | $70.0 \pm 2.4$ | $51.0 \pm 1.0$ | $\mathbf{68.2} \pm 2.7$ | $59.2 \pm 1.7$ | $66.9 \pm 3.4$ | $\mathbf{52.5} \pm 1.3$ |

Ravi & Larochelle, 2017) or FC100 (Oreshkin et al., 2018) (which is based on CIFAR100 (Krizhevsky, 2009)) and Mini-ImageNet. The order of appearance of two tasks within meta-training sequences is alternated for every batch. We compare systems trained with or without the backward transfer term in the ACL loss (the last term in Eq. 4).

Table 1 shows the results when the meta-trained models are meta-tested on the corresponding test sets of the few-shot learning datasets used for training. We observe that in both pairs of training tasks, the models without the ACL loss catastrophically forget the first task after learning the second one: the accuracy on the first task is at the chance level of about 20% for 5-way classification after learning the second task in-context. The ACL loss clearly addresses this problem: the CL algorithm learned through ACL preserves the performance of the first task. This effect is particularly pronounced in the Omniglot/Mini-ImageNet case (involving two rather distinct tasks).

Table 2 shows a similar evaluation but on two standard datasets, 5-way MNIST and CIFAR10. Again, ACL-trained models better preserve the memory of the first task after learning the second one. In the Omniglot/Mini-ImageNet case, we even observe certain positive backward tranfer effects: in particular, in the "MNIST-then-CIFAR10" continual learning case, the performance on MNIST noticeably improves after learning CIFAR10 (possibly leveraging 'more data' provided in-context).

## 4.2 In-Context Catastrophic Forgetting in the Baselines

Here we report some empirical observations on the baseline models trained **without** the backward transfer term (the last/third term in Eq. 4) in the ACL objective loss (corresponding to the **ACL/No** cases in Tables 1 and 2), namely emergence of "in-context catastrophic forgetting" during training. We focus on the Omniglot/Mini-ImageNet case, but similar trends can also be observed in the FC100/Mini-ImageNet case. Figures 2a and 2b show two cases we typically observe. These figures show an evolution of six individual loss terms (the lower the better), reported separately for the cases where Task A (here Omniglot) or Task B (here Mini-ImageNet) appears at the first (1) or second (2) position in the 2-task continual learning training sequences. 4 out of 6 curves correspond to the learning progress, showing whether the model becomes capable of in-context learning the given task (A or B) at the given position (1 or 2). The 2 remaining curves are the ACL backward tranfer loss, also measured for Task A and B separately here.

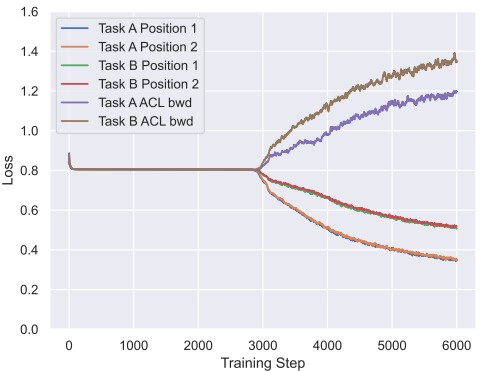 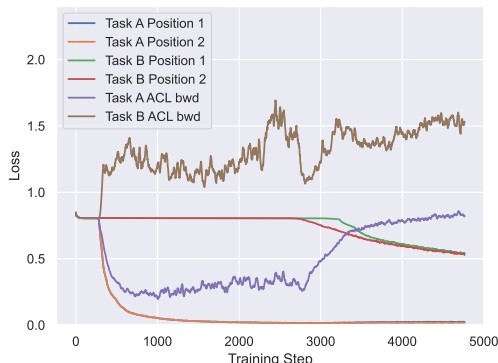

(a) Case where the model starts to learn two tasks simultaneously.

(b) Case where one task is learned first (here Task A) then the other one is learned later.

Figure 2: **ACL/No**-case meta-training curves displaying 6 individual loss terms, when the last term of the ACL objective (the backward tranfer loss; "*Task A ACL bwd*" and "*Task B ACL bwd*" in the legend) is **not** minimized (**ACL/No** case in Tables 1 and 2). Here Task A is Omniglot and Task B is Mini-ImageNet. We observe that, in both cases, without explicit minimization, backward transfer capability (*purple* and *brown* curves) of the learned learning algorithm gradually degrades as it learns to learn a new task (all other colors), causing in-context catastrophic forgetting. Note that *blue/orange* and *green/red* curve pairs almost overlap; indicating that when a task is learned, the model can learn it whether it is in the first or second segment of the continual learning sequence.

Figure 2a shows the case where two tasks are learned about at the same time. We observe that when the learning curves go down, the ACL losses go up, indicating that more the model learns, more it tends to forget the task in-context learned previously. This trend is similar when one task is learned before the other one as is the case in Figure 2b. Here Task A alone is learned first; while Task B is not learned, both learning and ACL curves go down for Task A (essentially, as the model does not learn the second task, there is no force that encourages forgetting). At around 3000 steps, the model also starts learning Task B. From this point, the ACL loss for Task A also starts to go up, indicating again a sort of opposing force effect between learning a new task and remembering a past task.

These observations clearly indicate that, without explicitly taking into account the backward transfer loss as part of learning objectives, the gradient descent search tends to find solutions/CL algorithms that prefer to erase previously learned knowledge (this is rather intuitive; it seems easier to find such algorithms that ignore any influence of the current learning to past learning than "better" ones that preserve prior knowledge). In all cases, we find our ACL objective to be crucial for the learned CL algorithms to be capable of remembering the old task while also learning the new task at the same.

## 4.3 GENERAL EVALUATION

**Evaluation on standard Split-MNIST.** Here we evaluate ACL on the standard Split-MNIST task in domain-incremental and class-incremental settings (Hsu et al., 2018; Van de Ven & Tolias, 2018), and compare its performance to existing CL and meta-CL algorithms (see Appendix A.6 for full references of these methods). Our comparison focuses on methods that do not require replay memory. Table 3 shows the results. Since our ACL-trained models are general-purpose learners, they can be directly evaluated (meta-tested) on a new task, here Split-MNIST. "ACL (Out-of-the-box model)" row of Table 3 corresponds to our model from Sec. 4.1 meta-trained on Omniglot and Mini-ImageNet using the 2-task ACL objective. It performs very competitively with the best existing methods in the domain-incremental setting, while it largely outperforms them in the 2-task class-incremental setting. The same model can be further meta-finetuned using the 5-task version of the ACL loss (here we only used Omniglot as the meta-training data). The resulting model (the last row of Table 3) outperforms all other methods in all settings studied here. We refer to Appendix A.6 for further discussions.

**Evaluation on diverse task domains.** Here we evaluate our ACL-trained models for CL involving more tasks/domains; using meta-test sequences made of MNIST, CIFAR-10, and Fashion MNIST. We also evaluate the impact of the number of tasks in the ACL objective: in addition to the model trained

Table 3: Classification accuracies (%) on the **Split-MNIST** domain-incremental and class-incremental learning (CIL) settings (Hsu et al., 2018). Both tasks are 5-task CL problems. For the CIL case, we also report the 2-task case for which we can directly evaluate our out-of-the-box ACL meta-learner of Sec. 4.1 (trained with a 5-way output and the 2-task ACL loss) which, however, is not applicable (N.A.) to the 5-task CIL requiring a 10-way output. Mean/std over 10 training/meta-testing runs. None of the methods here requires replay memory. See Appendix A.6 for further details.

| | Domain Incremental | Class Incremental | |
| Method | 5-task | 2-task | 5-task |
|---|---|---|---|
| Stochastic Gradient Descent (SGD) | $63.2 \pm 0.4$ | $48.8 \pm 0.1$ | $19.5 \pm 0.1$ |
| Adam | $55.2 \pm 1.4$ | $49.7 \pm 0.1$ | $19.7 \pm 0.1$ |
| Adam + L2 | $66.0 \pm 3.7$ | $51.8 \pm 1.9$ | $22.5 \pm 1.1$ |
| Elastic Weight Consolidation (EWC) | $58.9 \pm 2.6$ | $49.7 \pm 0.1$ | $19.8 \pm 0.1$ |
| Online EWC | $57.3 \pm 1.4$ | $49.7 \pm 0.1$ | $19.8 \pm 0.1$ |
| Synaptic Intelligence (SI) | $64.8 \pm 3.1$ | $49.4 \pm 0.2$ | $19.7 \pm 0.1$ |
| Memory Aware Synapses (MAS) | $68.6 \pm 6.9$ | $49.6 \pm 0.1$ | $19.5 \pm 0.3$ |
| Learning w/o Forgetting (LwF) | $71.0 \pm 1.3$ | - | $24.2 \pm 0.3$ |
| Online-aware Meta Learning (OML; Out-of-the-box) | $69.9 \pm 2.8$ | $46.6 \pm 7.2$ | $24.9 \pm 4.1$ |
| + optimized # meta-testing iterations | $73.6 \pm 5.3$ | $62.1 \pm 7.9$ | $34.2 \pm 4.6$ |
| ACL (Out-of-the-box model; Sec. 4.1) | $72.2 \pm 0.9$ | $71.5 \pm 5.9$ | N.A. |
| + meta-finetuned with 5-task ACL loss | $\mathbf{84.3} \pm 1.2$ | $\mathbf{93.4} \pm 1.2$ | $\mathbf{74.6} \pm 2.3$ |

on Omniglot/Mini-ImageNet using the 2-task ACL (Sec. 4.1), we also meta-train a model (with the same architecture and hyper-parameters) using 3 tasks, Omniglot, Mini-ImageNet, and FC100, using the 3-task ACL objective (see Appendix A.4). Note that the 3-task version is not only meta-trained for longer CL, but also meta-trained using more data. Table 4 shows the results. We observe that both ACL-trained models are indeed capable of retaining the knowledge without catastrophic forgetting for multiple tasks during meta-testing, while we also observe that the performance on prior tasks gradually degrade as the model learns new tasks. The 3-task version outperforms the 2-task one overall, encouragingly indicating a potential for further improvements even with a fixed parameter count.

## 5 DISCUSSION

**Prior work.** While there are several prior works that are catagorized as 'meta-continual learning' or 'continual meta-learning' (see, e.g., Javed & White (2019); Beaulieu et al. (2020); Caccia et al. (2020); He et al. (2019); Yap et al. (2021); Munkhdalai & Yu (2017), most of them are based on "model-agnostic meta-learning" (MAML; Finn et al. (2017); Finn & Levine (2018)) and learn *representations* for CL but still make use of classic CL algorithms. In particular, tuning of the learning rate and number of iterations is still required for optimal performance (see, e.g., Appendix A.6). In contrast, our approach learn *learning algorithms* in the spirit of Hochreiter et al. (2001); Younger et al. (1999); this may be categorized as 'in-context continual learning'. Several recent works (see, e.g., Irie & Schmidhuber (2023a); Coda-Forno et al. (2023); von Oswald et al. (2023b)) mention the possibility of such in-context CL without any concrete study. We show that in-context learning also suffers from catastrophic forgetting (Sec. 4.1-4.2) and propose ACL to address this problem.

**Artificial v. Natural ACL in Large Language Models?** Recently, "on-the-fly" or few-shot/meta learning capability of sequence processing NNs has attracted broader interests in the context of large language models (LLMs; Radford et al. (2019)). In fact, the task of language modeling itself has a form of *sequence processing with error feedback* (essential for meta-learning (Schmidhuber, 1990)): the correct label to be predicted is fed to the model with a delay of one time step in an auto-regressive manner. Trained on a large amount of text covering a wide variety of credit assignment paths, LLMs exhibit certain sequential few-shot learning capabilities in practice (Brown et al., 2020). This was rebranded as *in-context learning*, and has been the subject of numerous recent studies (e.g., Xie et al. (2022); Min et al. (2022); Yoo et al. (2022); Chan et al. (2022b;a); Kirsch et al. (2022); Akyürek et al. (2023); von Oswald et al. (2023a); Dai et al. (2023)). Here we explicitly/artificially construct ACL

Table 4: 5-way classification accuracies using 15 examples for each class for each task in the context. 2-task models are meta-trained on Omniglot and Mini-ImageNet, while 3-task models are in addition meta-trained on FC100. 'A, B' in 'Context/Train' column indicates that models sequentially observe meta-test training examples of Task A then B; evaluation is only done at the end of the sequence.

| Meta-Testing Tasks | | ACL Number of Meta-Training Tasks | |
|---|---|---|---|
| Context/Train | Test | 2 | 3 |
| A: MNIST-04 | A | $75.4 \pm 3.0$ | $89.7 \pm 1.6$ |
| B: CIFAR10-04 | B | $51.6 \pm 1.3$ | $55.3 \pm 0.9$ |
| C: MNIST-59 | C | $63.0 \pm 3.3$ | $76.1 \pm 2.0$ |
| D: FMNIST-04 | D | $54.8 \pm 1.3$ | $59.2 \pm 4.0$ |
| | Average | 61.2 | 70.1 |
| A, B | A | $81.5 \pm 2.7$ | $88.0 \pm 2.2$ |
| | B | $50.8 \pm 1.3$ | $52.9 \pm 1.2$ |
| | Average | 66.1 | 70.5 |
| A, B, C | A | $64.5 \pm 6.0$ | $82.2 \pm 1.7$ |
| | B | $50.8 \pm 1.2$ | $50.3 \pm 2.0$ |
| | C | $33.7 \pm 2.2$ | $44.3 \pm 3.0$ |
| | Average | 49.7 | 58.9 |
| A, B, C, D | A | $64.3 \pm 4.8$ | $78.9 \pm 2.3$ |
| | B | $47.5 \pm 1.0$ | $49.2 \pm 1.3$ |
| | C | $32.7 \pm 1.9$ | $45.4 \pm 3.9$ |
| | D | $31.2 \pm 4.9$ | $30.1 \pm 5.8$ |
| | Average | 43.9 | 50.9 |

training sequences and its objectives, but in modern large language models trained on a large amount of data mixing a large diversity of dependencies using a large backpropagation span, it is conceivable that some ACL-like objectives may naturally appear in the data.

**Limitations.** While we argue that our ACL is a promising approach for automating development of CL algorithms, there are also several limitations. First of all, directly scaling ACL for real-world tasks requiring many more classes does not seem straightforward: it would require very long training sequences. That said, it is also possible that ACL could be achieved without exactly following the process we propose; as we mention above for the case of LLMs, certain real-world data may naturally give rise to an ACL-like objective. Another limitation of this work is training within the limited span. It should be noted that unlike the standard Transformers, linear Transformers/FWPs like SRWMs can be trained by *carrying over states* across two consecutive batches for arbitrarily long sequences. Such an approach has been successfully applied to language modeling with FWPs (Schlag et al., 2021). This possibility, however, has not been investigated here, and is left for future work. This work is also limited to the task of image classification, which can be solved by feedforward NNs. Future work may investigate the possibility to extend ACL to continual learning of sequence learning tasks, such as continually learning new languages. Finally, ACL learns CL algorithms that are specific to the pre-specified model architecture; more general meta-learning algorithms may aim at achieving learning algorithms that are applicable to any model, as is the case for many classic learning algorithms.

## 6 CONCLUSION

Our novel Automated Continual Learning (ACL) formulates continual learning as sequence learning across long time lags, and trains sequence-processing self-referential neural networks (SRNNs) to learn their own in-context continual (meta-)learning algorithms. ACL encodes classic desiderata for continual learning (such as knowledge preservation, forward and backward transfer, etc.) into the objective function of the meta-learner. ACL uses gradient descent to deal with classic challenges of CL, to automatically discover CL algorithms with good behavior. Once trained, our SRNNs autonomously run their own continual learning algorithms without requiring any human intervention. Our experiments demonstrate the effectiveness and promise of the proposed approach.

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

## A    EXPERIMENTAL DETAILS

### A.1    DATASETS

For classic image classification datasets such as MNIST (LeCun et al., 1998), CIFAR10 (Krizhevsky, 2009), and FashionMNIST (FMNIST; Xiao et al. (2017)) we refer to the original references for details.

For Omniglot (Lake et al., 2015), we use Vinyals et al. (2016)'s 1028/172/432-split for the train/validation/test set, as well as their data augmentation methods using rotation of 90, 180, and 270 degrees. Original images are grayscale hand-written characters from 50 different alphabets. There are 1632 different classes with 20 examples for each class.

Mini-ImageNet contains color images from 100 classes with 600 examples for each class. We use the standard train/valid/test class splits of 64/16/20 following (Ravi & Larochelle, 2017).

FC100 is based on CIFAR100 (Krizhevsky, 2009). 100 color image classes (600 images per class, each of size $32 \times 32$) are split into train/valid/test classes of 60/20/20 (Oreshkin et al., 2018).

We use `torchmeta` (Deleu et al., 2019) which provides common few-shot/meta learning settings for these datasets to sample and construct their meta-train/test datasets.

### A.2    HYPER-PARAMETERS

We use the same model and training hyper-parameters in all our experiments. All hyper-parameters are summarized in Table 5. We use the Adam optimizer with the standard Transformer learning rate warmup scheduling (Vaswani et al., 2017). The vision backend is the classic 4-layer convolutional NN of Vinyals et al. (2016). Most configurations follow those of Irie et al. (2022c); except that we initialize the 'query' sub-matrix in the self-referential weight matrix using a normal distribution with a mean value of 0 and standard deviation of $0.01/\sqrt{d_{\text{head}}}$ while other sub-matrices use an std of $1/\sqrt{d_{\text{head}}}$ (motivated by the fact that a generated query vector is immediately multiplied with the same SRWM to produce a value vector). For any further details, we'll refer the readers to our public code we'll release upon acceptance.

Table 5: Hyper-parameters.

| Parameters | Values |
| --- | --- |
| Number of SRWM layers | 2 |
| Total hidden size | 256 |
| Feedforward block multiplier | 2 |
| Number of heads | 16 |
| Batch size | 16 or 32 |

### A.3    EVALUATION PROCEDURE

For evaluation on few-shot learning datasets (i.e., Omniglot, Mini-Imagenet and FC100), we use 5 different sets consisting of 32 K random test episodes each, and report mean and standard deviation.

For evaluation on standard datasets, we use 5 different random support sets for in-context learning, and evaluate on the entire test set. We report the corresponding mean and standard deviation across these 5 evaluation runs.

### A.4    THREE-TASK ACL

We can straightforwardly extend the 2-task version of ACL presented in Sec. 3 to more tasks. In the 3-task case (we denote the three tasks as $\mathbf{A}$, $\mathbf{B}$, and $\mathbf{C}$) used in Sec. 4.3, the objective function contains six terms. Following three terms are added to Eq. 4:

$$-\left(\log(p(y_{\text{test}}^{\mathcal{C}}|\boldsymbol{x}_{\text{test}}^{\mathcal{C}};\boldsymbol{W}_{\mathcal{A},\mathcal{B},\mathcal{C}})) + \log(p(y_{\text{test}}^{\mathcal{B}}|\boldsymbol{x}_{\text{test}}^{\mathcal{B}};\boldsymbol{W}_{\mathcal{A},\mathcal{B},\mathcal{C}})) + \log(p(y_{\text{test}}^{\mathcal{A}}|\boldsymbol{x}_{\text{test}}^{\mathcal{A}};\boldsymbol{W}_{\mathcal{A},\mathcal{B},\mathcal{C}}))\right) \quad (5)$$

## A.5 Auxiliary 1-shot Learning Objective

In practice, instead of training the models only for "15-shot learning," we also add an auxiliary loss for 1-shot learning. This naturally encourages the models to learn in-context from the first examples.

## A.6 Details of the Split-MNIST experiment

Here we provide details of the Split-MNIST experiments presented in Sec. 4 and Table 3. The baseline methods presented in Table 3 include: standard SGD and Adam optimizers, Adam with the L2 regularization, elastic weight consolidation (Kirkpatrick et al., 2017) and its online variant (Schwarz et al., 2018), synaptic intelligence (Zenke et al., 2017), memory aware synapses (Aljundi et al., 2018), learning without forgetting (LwF; Li & Hoiem (2016)). For these methods, we directly take the numbers reported in Hsu et al. (2018) for the 5-task domain/class-incremental settings.

For the 2-task class incremental setting, we use Hsu et al. (2018)'s code to train the correspond models (the number for LwF is currently missing as it is not implemented in their code base; we will add the corresponding/missing entry in Table 3 for the final version of this paper).

Finally we also evaluate a MAML-based meta-learning approach, OML (Javed & White, 2019). We note that as reported by Javed & White (2019) in their public code repository; after some critical bug fix, the performance of their OML matches that of Beaulieu et al. (2020) (which is a direct application of OML to another model architecture). Therefore, we focus on OML as our main meta-continual learning baseline. We take the out-of-the-box model (meta-trained for Omniglot, with a 1000-way output) made publicly available by (Javed & White, 2019). We evaluate the corresponding model in two ways. In the first, 'out-of-the-box' case, we take the meta-pre-trained model and only tune its meta-testing learning rate (which is done by (Javed & White, 2019) even for Omniglot meta-testing). We find that this method does not perform very well. In the other case ('optimized # meta-testing iterations'), we additionally tune the number of meta-test training iterations. We've done a grid search of the meta-test learning rate in $3 * \{1e^{-2}, 1e^{-3}, 1e^{-4}, 1e^{-5}\}$ and the number of meta-test training steps in $\{1, 2, 5, 8, 10\}$; we found the learning rate of $3e^{-4}$ and $8$ steps to consistently perform the best in all our settings. We've also tried it 'with' and 'without' the standard mean/std normalization of the MNIST dataset; better performance was achieved without such normalization (which is in fact consistent as they do not normalize the Omniglot dataset for their meta-training/testing). Their performance on the 5-task class-incremental setting is somewhat surprising/disappointing (since genenralization from Omniglot to MNIST is typically straightforward, at least, in common non-continual few-shot learning settings; see, e.g., Munkhdalai & Yu (2017)). At the same time, to the best of our knowledge, OML-trained models have not been tested in such a condition in prior work; from what we observe, the publicly available out-of-the-box model might be overtuned for Omniglot/Mini-ImageNet or the frozen 'representation network' is not ideal for genenralization.

Unlike any other methods above, our out-of-the-box ACL models (trained on Omniglot and Mini-ImageNet) do not require any tuning at meta-test time. Nevertheless, we've checked the effect of the number of meta-test training examples (5 vs. 15; 15 is the number used using meta-training); we found 15 examples to work better. For the version that is meta-finetuned using the 5-task ACL objective (using only the Omniglot dataset), we use 5 examples for both meta-train and meta-test training.

