# OpenReview forum: "Automating Continual Learning"
_ICLR.cc/2024/Conference — Submitted to ICLR 2024_

### Official Review · Reviewer_StyD · 2023-10-28

**Soundness:** 3 good
**Presentation:** 2 fair
**Contribution:** 2 fair
**Rating:** 5
**Confidence:** 4

**Summary:**

This paper proposes to formulate continual learning as a sequence learning problem and applies self-referential weight matrices (SRWM), which can be considered a sequence model, as the key mechanism for continual learning.
SRWM is a linear layer that produces self-modification as an auxiliary output.

**Strengths:**

I agree with the general direction of this paper that formulates continual learning as a sequence learning problem.
This idea of formulating a learning process as sequence learning has been used in the meta-learning literature, especially for few-shot settings, but has not been utilized in the continual learning domain.
I believe this direction requires further investigation.

**Weaknesses:**

### Missing Related Works in Meta-Continual Learning

According to my understanding, this work should be classified as a meta-continual learning (MCL) approach, which is also referred to as *learning to continually learn*.
It is a direct extension of meta-learning that replaces each learning episode with a continual learning episode, which also aligns with the authors' description in section 2.2.
There are several important prior works in this domain [1, 2, 3] that were not mentioned in the paper.
They should also be compared as baselines.

### Confusing Description About the Experimental Settings

Since MCL is a branch of meta-learning that aims to optimize a learning algorithm, it is crucial to separate meta-training and meta-test sets.
Also, there should be no overlap in the constituent tasks between them.
Otherwise, the model can achieve a high score simply by memorizing the tasks in the meta-training set without learning new knowledge during the meta-test phase.

In the paper, I could not find any description of how meta-training and meta-test sets are constructed.
I suspected that the authors would have followed the conventional meta-splits for Omniglot and Mini-ImageNet datasets, but I suggest the authors refine the overall terminology to be consistent with the existing meta-learning or MCL literature.

### Weak Experimental Results

The proposed method is tested only on two-task and three-task CL scenarios, which is an unreasonably tiny scale compared to previous works on MCL [1, 2, 3].
I do not think such a small number of tasks can be considered meaningful.

### Reproducibility

It seems hard to reproduce the experimental results solely from the provided text.
To verify and reproduce experimental results, I believe that including code with the submission should be the standard practice.

---
- [1] Javed, Khurram and Martha White. “Meta-Learning Representations for Continual Learning.” NeurIPS (2019).
- [2] Beaulieu, Shawn L. E. et al. “Learning to Continually Learn.” ECAI (2020).
- [3] Banayeeanzade, Mohammadamin et al. “Generative vs. Discriminative: Rethinking The Meta-Continual Learning.” NeurIPS (2021).

**Questions:**

- Experiments with much longer training sequences, as in [1, 2, 3], seem necessary.
- Since each convolution filter is just a linear layer applied to a local patch, shouldn't it be possible to construct a CNN version of ACL?
- I have doubts about the representational capability of SRWM since the complex learning dynamics in non-stationary streams depend solely on the initial parameters. Is it really sufficient to manipulate the initial parameters? Can SRWM really handle long sequences?

---

> ### Author Response · Authors · 2023-11-21
> **Response to Reviewer StyD (part 1/2)**
>
> **NB: Please find our response to other questions in "Common Responses to All Reviewers" above. Thank you.**
>
> We thank the reviewer for valuable time reviewing our work and for valuable comments.
>
> > There are several important prior works in this domain [1, 2, 3] that were not mentioned in the paper. They should also be compared as baselines.
>
> The reviewer is right: these citations were missing. We actually had realized this in the meanwhile, so we had already added these MAML-based/like approaches in our internally updated paper.
> Following the reviewer’s suggestion, the updated paper includes a comparison to [1] on Split-MNIST (Table 3; the experimental details can be found in Appendix A.6). Please note that according to the readme file in the official repository of [1] https://github.com/khurramjaved96/mrcl, *“... it is possible to get the same results as ANML (S. Beaulieu 2020) without using any neuromodulation layers”* We concluded that comparing to [2] is unnecessary here; we focus on [1] as our meta-continual learning baseline.
>
> That said, regarding:
>
> > According to my understanding, this work should be classified as a meta-continual learning (MCL) approach, which is also referred to as learning to continually learn.
>
> While our method can definitely be categorized as a meta-learning approach for continual learning, it should not be categorized as “learning to continually learn” in the sense of [2], since ACL learns a learning algorithm while MAML based methods [1, 2] only learn *representations* for CL (or representation learning networks) and still relies on the standard learning algorithm (typically Adam). These are fundamentally different methods even if the title of [2] is very broad, and gives the impression of generality; that is not the case. In particular, their meta-testing requires tuning the learning rate—and in addition, in our experiments/Table 3/Appendix A.6, we also observed that they also require tuning meta-test *training iterations* to perform well on an unseen domain, (Split-)MNIST.
> In contrast, our learned learning algorithm, once trained, does not require any tuning (learning rate is self-regulated by the trained SRWM; sigma(beta) in Eq.3) This distinction is important. We clarify this in the updated “prior work” paragraph (Page 8, Sec 5.).
>
>
> > Since MCL is a branch of meta-learning that aims to optimize a learning algorithm, it is crucial to separate meta-training and meta-test sets. Also, there should be no overlap in the constituent tasks between them. Otherwise, the model can achieve a high score simply by memorizing the tasks in the meta-training set without learning new knowledge during the meta-test phase.
>
> Yes, of course, we are aware of this. We used the standard few-shot learning sequence construction process to ensure this (see further comment in the next point). Otherwise, it is anyway impossible to obtain a model that can learn unseen MNIST or CIFAR-10 during meta-testing, while only meta-trained on Omniglot and Mini-ImageNet (see Table 2).
>
> > In the paper, I could not find any description of how meta-training and meta-test sets are constructed. I suspected that the authors would have followed the conventional meta-splits for Omniglot and Mini-ImageNet datasets,
>
> Yes, absolutely. In the Appendix A.1., we had mentioned that we used `torchmeta` for that, but we agree this was not clear in the main text (relating to the next point).
>
> > but I suggest the authors refine the overall terminology to be consistent with the existing meta-learning or MCL literature.
>
> The reviewer is right; our terminology was confusing (which likely caused Reviewer jN2L’s confusion). Following this suggestion, we now use “meta-training train/test” and “meta-test train/test” terminology in the updated paper. Thank you for pointing this out.
>
> > The proposed method is tested only on two-task and three-task CL scenarios,
>
> This is not accurate; while we had only used 2-task and 3-task objective functions, we evaluated our models on 4-task CL (now called Table 4). Now with Split-MNIST, we also test our method for 5-task settings.

---

> > ### Author Response · Authors · 2023-11-21
> > **Response to Reviewer StyD (part 2/2)**
> >
> > > which is an unreasonably tiny scale compared to previous works on MCL [1, 2, 3]. I do not think such a small number of tasks can be considered meaningful.
> >
> > > Experiments with much longer training sequences, as in [1, 2, 3], seem necessary.
> >
> > We’d like to note that focusing only on the number of tasks is a restricted view on the problem of CL and to evaluate significance of CL methods. [1, 2] focus on the number of tasks but ignore the importance of task diversity (e.g., they only meta-train on Omniglot and meta-test on Omniglot). Domain generalization is another important aspect (which is classic in the context of few-shot learning; see e.g., Munkhdalai and Yu ICML 2017 “Meta networks” we cited) and that also impacts the way we design algorithms. This seems particularly relevant in the CL setting where we want the system to continually learn new tasks. Effectively, when tested on an unseen domain, (Split-)MNIST, OML [1]’s out-of-the-box model performance is limited even with 5 tasks (see Table 3; we provide further comment on this in Appendix A.6). While we agree that scaling our method to many more tasks is a very important direction, we argue that our current set of experiments (now that we also included the standard 5-task Split-MNIST settings thanks to the reviewers’ suggestions) successfully demonstrates the promise of our new method.
> >
> >
> > > It seems hard to reproduce the experimental results solely from the provided text. To verify and reproduce experimental results, I believe that including code with the submission should be the standard practice.
> >
> > We fully agree with the reviewer on the importance of public code; in fact, we had mentioned on page 1 (footnote 1) that we’ll release the code in a public repository upon acceptance (as well as in Sec 3). Since this promise did not seem to be enough to satisfy the reviewer, we’ll upload the code as supplemental material (sorry, please wait until tomorrow). Upon acceptance, we’ll also release our pre-trained models.
> >
> > As a side note, we made use of the public code of Javed and White 2019 and Hsu et al. 2018 for our Split-MNIST experiments; we will acknowledge this in the final version.
> >
> > > Since each convolution filter is just a linear layer applied to a local patch, shouldn't it be possible to construct a CNN version of ACL?
> >
> > Please be note that we need *efficient implementation* for any of these experiments. Adding a self-referential logic into a highly optimized CUDA implementation of convolution is technically non-trivial (except maybe for some professional CUDA programmer; but we’d for sure not call it “just a linear layer” in this sense). We instead discuss MLP-mixer in Sec 3. because it is much more approachable as they are typically implemented using regular PyTorch linear layers (which can be replaced by our SRWM CUDA kernel); that said, as it turns out, MLP-mixer+SRWM also still remains too slow with such a lazy implementation.
> >
> > > I have doubts about the representational capability of SRWM since the complex learning dynamics in non-stationary streams depend solely on the initial parameters. Is it really sufficient to manipulate the initial parameters? Can SRWM really handle long sequences?
> >
> > We respectfully note that the reviewer is missing one basic point here. We learn “complex dynamics” using solely “initial parameters” all the time. Please think of any (standard) sequence processing NNs (LSTM, Transformers); they only have their “initial” weight matrices to process sequences (which can have arbitrary complex dynamics). SRWMs are similar except that they can also modify their “initial” weights over time (unlike conventional models that process the entire sequences with frozen/fixed weights).
> > Regarding the length generalization, the question is again similar to the one for any other sequence processors (maybe linear Transformers in particular, due to their relation; Sec 2.3), it can generalize to a certain degree, but longer generalization requires certain training tricks (we discuss this in the “limitations” paragraph/Sec 5./Page 9); which is not specific to SRWM.
> >
> >
> > We hope our response successfully addresses the reviewer’s main concerns. We believe the current/updated version of the paper demonstrates that the proposed method is an interesting new approach for continual learning.  If you find our response useful/convincing, please consider increasing the score. Thank you very much.

---

> > > ### Author Response · Authors · 2023-11-21
> > >
> > > Since the reviewer requested, we've uploaded our code as supplemental material to prove our intention of releasing official code upon acceptance (which we had promised in the original submission).

---

> > > > ### Comment · Reviewer_StyD · 2023-11-22
> > > >
> > > > Thank you for the detailed response.
> > > > Many of my concerns are addressed, especially regarding writing and providing code.
> > > > However, several issues still remain.
> > > >
> > > > > "Meta-learning approach for continual learning" vs. "learning to continually learn"
> > > >
> > > > I think these two are basically the same.
> > > > As meta-learning is often referred to as "learning to learn," meta-continual learning (MCL) can be called "learning to continually learn."
> > > > In meta-learning, each learning episode is an offline learning episode, while in MCL, it is a CL episode.
> > > > Learning initializations is just one approach to MCL, not its definition (the title of [2] is indeed excessively general, and I think it is not a proper title).
> > > > Even in the meta-learning literature, some methods are based on MAML, while others are not.
> > > >
> > > >
> > > > > We’d like to note that focusing only on the number of tasks is a restricted view on the problem of CL and to evaluate significance of CL methods.
> > > >
> > > > This is correct.
> > > > Likewise, focusing solely on task diversity is also a restricted view.
> > > > Therefore, I expected an evaluation with a much larger number of tasks.
> > > > If not hundreds as in [1, 2, 3], I anticipated at least dozens of tasks.
> > > > Even if the proposed method does not work well in such settings, I think it can provide valuable insights into the proposed approach.
> > > >
> > > > > [1, 2] focus on the number of tasks but ignore the importance of task diversity (e.g., they only meta-train on Omniglot and meta-test on Omniglot).
> > > >
> > > > Meta-training and meta-test sets sharing the same task distribution is one of the key assumptions in meta-learning [4] and MCL.
> > > > If we meta-train on Omniglot, we generally should not expect the learning algorithm to work well on Mini-ImageNet.
> > > > It is the same as training a classifier on MNIST and testing it on CIFAR-10 in a standard offline learning setting.
> > > > Generalizing to such out-of-distribution (meta-)test data is an orthogonal research direction.
> > > >
> > > >
> > > > [4] T. Hospedales, A. Antoniou, P. Micaelli and A. Storkey, "Meta-Learning in Neural Networks: A Survey" in IEEE Transactions on Pattern Analysis & Machine Intelligence, vol. 44, no. 09, pp. 5149-5169, 2022.

---

> > > > > ### Author Response · Authors · 2023-11-22
> > > > >
> > > > > We really appreciate your reply. Thank you very much.
> > > > >
> > > > > > Many of my concerns are addressed, especially regarding writing and providing code.
> > > > >
> > > > > We are glad to read that these concerns have been resolved; in particular in light of ongoing discussion with Reviewer jN2L about writing...
> > > > >
> > > > > > "Meta-learning approach for continual learning" vs. "learning to continually learn"
> > > > > > I think these two are basically the same.
> > > > >
> > > > > Yes, we fully agree with this statement. To clarify, we did not disagree on this at all: what we wrote was that "learning to continually learn **in the sense of [2]**" is not the same as ours for the reason we explained; it is only a subset of "learning to continually learn" (as the reviewer has also pointed out in the reply).
> > > > >
> > > > > >  Likewise, focusing solely on task diversity is also a restricted view
> > > > >
> > > > > We fully agree with this too. That said, now that we have the 5-task Split-MNIST benchmark, it seems to us that we reached the "threshold" where it can be regarded as an "acceptable" CL setting.
> > > > > Also, somehow no reviewer has reacted to the results we obtained on Split-MNIST but they look really good among methods that do not require replay memory or parameter increase. If the reviewer is aware of better methods under this condition, please let us know the references.
> > > > >
> > > > > > If not hundreds as in [1, 2, 3], I anticipated at least dozens of tasks. Even if the proposed method does not work well in such settings, I think it can provide valuable insights into the proposed approach.
> > > > >
> > > > > We actually do not see any conceptual reasons our model would not work on more tasks but it would take more compute/time; we won't be able to show this during the rebuttal, but we can promise to run a 10-task setting for the final version, in order to provide more "insights into the proposed approach" as requested by the reviewer. What do you think?
> > > > >
> > > > > > Meta-training and meta-test sets sharing the same task distribution is one of the key assumptions in meta-learning [4] and MCL. If we meta-train on Omniglot, we generally should not expect the learning algorithm to work well on Mini-ImageNet.
> > > > >
> > > > > We partially disagree. We agree that the specific example the reviewer brought is a hard one (meta-train on Omniglot, and meta-test on Mini-ImageNet) but the opposite should work (meta-train on Mini-ImageNet, and meta-test on Omniglot) assuming that no "ill-behaving" components are used in the model architecture (such as batch normalization that heavily "overfits" to training task distributions). In fact, in the few-shot learning literature, generalization evaluation on unseen datasets is common: an old one is [5] which meta-train on Omniglot, and meta-test on MNIST (which works out of the box; as is the case with many MNIST families); in multi-task settings, [6, 7] meta-train on 8 datasets (ILSVRC, Omniglot, etc...) and meta-test on 5 unseen datasets (MSCOCO, MNIST, CIFAR-10, ...). There is also "meta dataset" [8] that has a "Training-on-ImageNet-only" track.
> > > > >
> > > > > Overall, we are not sure to understand why the reviewer considers this perspective as an "issue".
> > > > > In contrast, we find this aspect particularly relevant and important for CL whose very goal is to continually learn new tasks.
> > > > >
> > > > > [5] Munkhdalai and Yu. ICML 2017. "Meta Networks". https://arxiv.org/abs/1703.00837
> > > > >
> > > > > [6] Bronskill et al. ICML 2020 "TaskNorm: Rethinking Batch Normalization for Meta-Learning". https://arxiv.org/abs/2003.03284
> > > > >
> > > > > [7] Requeima et al. NeurIPS 2019 "Fast and Flexible Multi-Task Classification Using Conditional Neural Adaptive Processes". https://arxiv.org/abs/1906.07697
> > > > >
> > > > > [8] https://github.com/google-research/meta-dataset
> > > > >
> > > > > > It is the same as training a classifier on MNIST and testing it on CIFAR-10 in a standard offline learning setting.
> > > > >
> > > > > We disagree with this. A regular classifier trained on MNIST has 0 chance to work on CIFAR-10 out of the box because there is no way for the model to know the underlying "input-class to output-label" mapping. In contrast, a meta-learning neural net has a chance to work as it takes both an input image and its correct label (of meta-test training examples) as its net inputs during meta-test training, and it has been meta-trained to learn from given examples. These are two different settings that are not comparable (but here we are really diverging from our original discussion about our submission).
> > > > >
> > > > > Overall, while the reviewer wrote that "several issues still remain", we only see one remaining concern (please correct us if we are wrong): we test only up to 5-task-long CL. Our main response to this is what we wrote above; we'd add that, in our view, the reviewer's argument would be only fully valid if the 5-task setting is completely solved by the existing MCL methods, and it is obsolete; this does not seem to be the case. Please let us know if the reviewer still thinks s/he has strong reasons to rate our work as borderline reject. Thank you very much.

---

> > > > > > ### Comment · Reviewer_StyD · 2023-11-22
> > > > > >
> > > > > > I apologize for not being entirely clear in the last comment since I was in a bit of a rush.
> > > > > > More details are provided in the following.
> > > > > >
> > > > > > To summarize, my main concern is that the experiments in this work still do not provide meaningful comparisons with the baselines.
> > > > > >
> > > > > > The primary goal of any learning algorithm (not just meta-learning approaches) is to perform well on what it learned in the training phase.
> > > > > > Anything other than that, such as out-of-distribution (OOD) generalization, is secondary.
> > > > > > I'm not saying that OOD generalization is useless; of course, it would be nice to have a good performance on OOD data, and there have been numerous works on OOD generalization.
> > > > > > What I want to emphasize is that a learning algorithm should have a good performance on in-distribution (ID) data first.
> > > > > >
> > > > > > Assume two learning algorithms, A and B.
> > > > > > Each algorithm produces a model trained on the same training set.
> > > > > > There are two types of test sets; one is an ID test set (sampled from the same distribution as the training set), and the other is an OOD test set.
> > > > > > If A scores better with the ID test set and B scores better with the OOD test set, which algorithm is better?
> > > > > > Generally, the answer would be A.
> > > > > > If the ID score is lower, the OOD score is mostly due to a lucky OOD test set; it can perform much worse in other OOD test sets.
> > > > > > Moreover, if an algorithm performs better on an ID test set, it is relatively easy to improve the training set by increasing the amount and scope of the data.
> > > > > >
> > > > > > The same principle applies to meta-learning and MCL at the meta-level: you should prioritize evaluation with an ID meta-test set (sharing the same task distribution as the meta-training set).
> > > > > > Even if the proposed method works better on a few OOD meta-test sets, it cannot be considered superior if it performs worse on ID meta-test sets.
> > > > > > For this reason, all the previous works on MCL [1, 2, 3] primarily use ID meta-test sets.
> > > > > >
> > > > > > While [1] is added as a baseline in the updated draft, only the scores of an OOD meta-test set, i.e., Split-MNIST, are reported after meta-training on Omniglot + Mini-ImageNet.
> > > > > > I think the authors should mainly report the ID scores.
> > > > > > The OOD scores are also not impressive.
> > > > > > It seems unreasonable to claim successful OOD generalization with merely 74.6% accuracy in MNIST.
> > > > > > This score may seem strong compared to other CL baselines, but other MCL approaches achieve far better accuracy in much more challenging scenarios if an ID meta-test set is used [1, 2, 3].
> > > > > >
> > > > > > Lastly, I think [3] should also be compared as a baseline.
> > > > > > It seems reasonable to skip [2] since [2] is an extension of [1] and does not seem to perform better than [1].
> > > > > > However, [3] is a completely different approach and is reported to perform significantly better than [1].
> > > > > > It may show stronger performance even in OOD meta-test sets.

---

> ### Author Response · Authors · 2023-11-22
>
> Thank you very much for these additional comments.
> We really appreciate the reviewer's engagement.
>
> We understood the reviewer's point on the ID evaluation. That said, regarding:
>
> > It seems unreasonable to claim successful OOD generalization with merely 74.6% accuracy in MNIST. This score may seem strong compared to other CL baselines, but other MCL approaches achieve far better accuracy in much more challenging scenarios if an ID meta-test set is used [1, 2, 3].
>
> We can not agree with this argument/reasoning.
> Our high-level goal/claim is to replace hand-crafted CL algorithms by a "better" learned CL algorithm.
> On Split-MNIST (a standard CL benchmark suggested by the two other reviewers), our learned algorithm effectively outperforms other hand-crafted CL baselines: "74.6% accuracy" has to be compared to the performance of the baselines which is around 20-25%, strongly supporting our original claim.
>
> We can not directly compare these numbers to other numbers achieved on other datasets which have nothing to do with this benchmark.
> We emphasize that we ran this Split-MNIST experiment not because it's our "lucky OOD test set" (in Table 2 and 4, we also evaluate our system on CIFAR-10 and F-MNIST) but because it is a standard/universal CL benchmark that the CL community is generally interested in (as suggested by the two other reviewers).
>
> Now we can add MCL baselines to this comparison; we already reported [1] (implicitly covering [2]) as requested by the reviewer; we could also add [3] or any other existing MCL methods, but that would not fundamentally diminish our contributions much since none of [1,2,3] really learns learning algorithms. As we explained in A.6., [1] is sensitive to meta-test hyperparameters tuning (learning rate, and number of iterations) which is still a characteristic of handcrafted algorithms---something we want to avoid to "automate CL." [3] indeed does not require tuning meta-test hyper-parameters, but it is based on a very different paradigm (of generative classifiers). These MCL methods are relevant (we agree) but they are not solutions to our original goal of replacing hand-crafted CL algorithms for neural nets. Our method is conceptually unique/new in this regard (also directly relevant to the now-popular in-context learning) and its effectiveness is clearly shown on a well-known standard benchmark; while this may still be experimentally limited, follow-up works may further improve this approach and apply it to other datasets... We'll respectfully leave the reviewers and AC to ultimately judge.
>
> Once again, we thank the reviewer for his/her engagement during this rebuttal, and for very useful comments.

---

### Official Review · Reviewer_jN2L · 2023-10-31

**Soundness:** 2 fair
**Presentation:** 3 good
**Contribution:** 2 fair
**Rating:** 5
**Confidence:** 4

**Summary:**

The paper proposes a new way to think about, and potentially solve, the continual learning problem (in particular, the supervised task incremental learning variation of CL).
This new approach views CL as a sequence learning problem. Each sub-sequence consists of input/target examples corresponding to one task to be learned. These sub-sequences can then form longer sequences, for multiple tasks, by concatenating multiple sub-sequences.
Once formulated as such a sequence-learning task, a gradient descent search for CL learning algorithms can look for the desired CL behavior by constructing loss functions that avoid catastrophic forgetting and aim to achieve goals such as forward transfer.

**Strengths:**

On the positive side, the approach of viewing the CL problem in the context of sequence learning seems interesting -- and it gives a fresh perspective to an area (CL) that is becoming increasingly incremental.

**Weaknesses:**

I have some major concerns about whether this method is actually doing CL (versus multitask learning).

Another major concern is whether ACL can be used in a practical context in which many tasks will be learned over time (as opposed to just a handful).

Please see comments below.

**Questions:**

On Page 6, the paper states: “Unless otherwise indicated, we concatenate 15 examples for each class for each task in the context during both training and evaluation (resulting in sequences of length 75 for each task).”
Having a temporally structured input during evaluation is not a valid approach in the context of CL (although I am aware that some meta-learning papers unfortunately do that -- but that does not mean that their approach can be accepted without question because it has been previously published). Such temporally structured inputs makes discrimination between classes of different tasks trivial. For example, if you give someone 75 Omniglot examples and 75 Imagenet examples and ask them to classify an input x during testing, I can easily determine whether x is from Omniglot or Imagenet without learning (just by computing some statistics of pixel values). Letters would, of course, look different than natural images. Then, predictions become much easier.

On Page 6, the paper states: “The order of appearance of two tasks within training sequences is alternated for every batch.” This sounds like both datasets are available at the same time. If that is true, what the paper is actually doing multitask learning, not CL.

Looking at the loss function (Equation 4), the first term requires access to old model weights W_A (linear growth in memory requirement as they see more tasks), the second term is okay in terms of CL, but the third term requires access to a previous test dataset, which violates CL. It may be that these are some form of “replay” examples, but the paper does not mention that.

I see that the method is significantly different from other continual learning methods, still I would expect the authors to benchmark against some existing methods. After all, the claim is that instead of hand-crafting CL algorithms, we can learn how to sequentially learn. Does ACL perform better than handcrafted tricks? The method is computation intensive, and it does not seem easily scalable to more tasks. So, I would at least want to see the paper outperform some existing methods in the two-task scenario to argue that learning how to continual learn is a promising direction to pursue.

If you examine Equation 5 on the last page, you'll notice that in order to learn a third task, they need to add three terms to the loss function. In continual learning, a five-task setting is considered small. To learn SplitMNIST, for example, they would actually need 1 + 2 + 3 + 4 + 5 (15) terms in the loss function. As a result, their method becomes quadratically more expensive in terms of computation (i.e., for Task n, you require backpropagation through (n)(n-1)/2 terms). This is clearly not practical.

---

> ### Author Response · Authors · 2023-11-21
> **Response to Reviewer jN2L (part 1/2)**
>
> **NB: Please find our response to other questions in "Common Responses to All Reviewers" above. Thank you.**
>
>
> We thank the reviewer for valuable time reviewing our work and for encouraging comments on the originality of this work (*“a fresh perspective to an area (CL) that is becoming increasingly incremental”*).
>
> The reviewer has both some critical misunderstandings of certain aspects of our method, AND really important/relevant comments at the same time. We’d like to resolve/respond to them all here.
>
> > I have some major concerns about whether this method is actually doing CL (versus multitask learning).
>
> [NB: We assumed that by “multitask learning” the reviewer means “joint training”.]
>
> Here the reviewer has some critical misunderstandings about our method. As we’ll explain, our framework is a proper CL framework. We are not affected by ANY of the problems/concerns raised by the reviewer regarding whether our method respects the framework of CL.
> We believe what causes some of these confusions is the meta-learning procedure, and our terminologies were not very helpful in that regard. To improve this, we introduce meta-training meta-testing terminology (as suggested by Reviewer StyD).
> Please let us try to resolve this confusion one-by-one as follows.
>
> > Having a temporally structured input during evaluation is not a valid approach in the context of CL (although I am aware that some meta-learning papers unfortunately do that -- but that does not mean that their approach can be accepted without question because it has been previously published). Such temporally structured inputs makes discrimination between classes of different tasks trivial. For example, if you give someone 75 Omniglot examples and 75 Imagenet examples and ask them to classify an input x during testing, I can easily determine whether x is from Omniglot or Imagenet without learning (just by computing some statistics of pixel values). Letters would, of course, look different than natural images. Then, predictions become much easier.
>
> [This confusion is independent of meta-learning]
> We first would like to clarify that all our main CL settings are “domain-incremental” (except the “class-incremental” setting in our new Split-MNIST experiments). This means that if our CL task consists of 2 tasks with both of them being 5-way classification, the output dimension of the model is also 5 (not 10), which is shared for both tasks. In this setting, the temporal structure can not help the model for classification: (taking the reviewer’s example) an input image has to be classified to be one of the 5 labels among {0, 1, 2, 3, 4} regardless of whether the image comes from Omniglot or Mini-ImageNet. A model may easily recognize that an input image is from Omniglot, that’s likely the case, but that would not facilitate the task; it still has to do the prediction among the 5 labels.

---

> > ### Author Response · Authors · 2023-11-21
> > **Response to Reviewer jN2L (part 2/2)**
> >
> > > On Page 6, the paper states: “The order of appearance of two tasks within training sequences is alternated for every batch.” This sounds like both datasets are available at the same time. If that is true, what the paper is actually doing multitask learning, not CL.
> >
> > This seems to be a misunderstanding related to the meta-learning procedure.
> > We believe that the best way to explain this is to first clarify the setting at a higher level.
> > Let’s say our goal is to learn two tasks sequentially: MNIST, then CIFAR-10 (as in our experiments of Table 2). Our ACL model does not make use of *any* MNIST or CIFAR-10 datasets during “meta-training”. The actual MNIST-to-CIFAR-10 continual learning happens, once the model is done with meta-training, during the process called “meta-testing”. During meta-testing, our model reads a sequence of input/label example pairs (called “meta-test training” examples) sampled from the training set of MNIST, then (similarly) a sequence of CIFAR-10 training examples; our model ends up in a weight state W_{MNIST, CIFAR-10} at the end of the entire sequence. No prediction has been done so far. Now using the resulting weights W_{MNIST, CIFAR-10}, we make predictions on examples (called “meta-test test” examples) from the test set of MNIST and CIFAR-10 to evaluate the model’s capability to predict both the first task, MNIST, and the second one, CIFAR-10.
> >
> > Now during meta-training (during which we train the learnable parameters of the model using gradient descent), the model has to be (meta-)trained to learn to solve a task from a sequence of examples (i.e., as is done during meta-testing described above). For that, each meta-training sequence has to represent/simulate a new, unknown, task that the model has never seen, so that it has to make use of provided “meta-training training” examples to solve the task; otherwise the model can just memorize the input/label mapping and becomes capable of making predictions while ignoring the examples provided in context. The construction of “meta-training training” sequences for a $N$-way classification, using a dataset containing $C$ classes works as follows; for each sequence, we sample $N$ random but distinct classes out of $C$ ($N < C$). The resulting $N$ classes are re-labelled such that each class is assigned to one out of $N$ distinct **random** label index which is unique to the sequence. Each such a sequence “simulates” an unknown task the model has to learn. In the case of ACL with the 2-task loss, we need two such sequences representing two tasks to be learned sequentially. In our experiments for Table 2/Top-part, we use Omniglot and Mini-ImageNet. We essentially sample such a meta-training sequence, one from each of the two datasets, say O and M, and concatenate them to form a single meta-training sequence [O, M]. Now, finally back to the original question; we could also concatenate these sequences in another order to get [M, O] instead. This is what we mean by “alternating” the order for every batch. As you should be able to notice now, this process has no impact on the “joint training vs. continual learning” question on the final MNIST-to-CIFAR-10 meta-testing we ultimately care about.
> >
> > We note that apart from the use of multiple tasks, this is the very standard procedure used in few-shot learning with sequence processing networks.
> > If the reviewer still finds anything confusing or has suggestions on improving the clarity, we’ll be happy to discuss more.
> >
> > > Looking at the loss function (Equation 4), the first term requires access to old model weights W_A (linear growth in memory requirement as they see more tasks), the second term is okay in terms of CL, but the third term requires access to a previous test dataset, which violates CL. It may be that these are some form of “replay” examples, but the paper does not mention that.
> >
> > Continuing from our description above, the loss function (Eq 4.) is used in the meta-training process. The meta-testing (the actual CL) is done without any (explicit) objective function (it’s just a forward pass of the model over the meta-test training examples to let the SRWM self-modify its own weights). There is no violation of the CL setting.
> >
> > We hope that these clarifications help. Regarding the original concern regarding missing comparisons, we believe our results on Split-MNIST provide a good overview and demonstrate the promise of ACL.
> > If the reviewer finds our response useful/convincing, please consider increasing the score. Thank you very much.

---

> > > ### Comment · Reviewer_jN2L · 2023-11-21
> > > **Response to authors**
> > >
> > > First, let me thank the authors for their detailed response to all reviews.
> > >
> > > Unfortunately, these responses did not address my concerns on whether the paper actually solves the Continual Learning problem -- at least in any practical manifestation of that problem. The following sentence, copied from the authors' response, shows how confusing the proposed approach is, for instance in terms of whether it uses training data during testing (but there are also other similarly confusing points):
> > >
> > > "During meta-testing, our model reads a sequence of input/label example pairs (called “meta-test training” examples) sampled from the training set of MNIST, then (similarly) a sequence of CIFAR-10 training examples; our model ends up in a weight state W_{MNIST, CIFAR-10} at the end of the entire sequence. No prediction has been done so far. Now using the resulting weights W_{MNIST, CIFAR-10}, we make predictions on examples (called “meta-test test” examples) from the test set of MNIST and CIFAR-10 to evaluate the model’s capability to predict both the first task, MNIST, and the second one, CIFAR-10."
> > >
> > > The fact that three independent reviewers had very similar concerns about the proposed approach means something in my opinion. I think that it would be beneficial for the paper if the authors largely re-write the paper, at least the problem formulation, the description of the method, and the setup of the experiments, also following the more common terminology and assumptions in the CL and metalearning literature.

---

> ### Author Response · Authors · 2023-11-21
>
> Thank you very much for your prompt response.
>
> > Unfortunately, these responses did not address my concerns on whether the paper actually solves the Continual Learning problem -- at least in any practical manifestation of that problem.
>
> We are sorry to read this. Please let us ask a few questions that might help.
>
> > The following sentence, copied from the authors' response, shows how confusing the proposed approach is, for instance in terms of whether it uses training data during testing (but there are also other similarly confusing points):
> "During meta-testing, our model reads a sequence of input/label example pairs (called “meta-test training” examples) sampled from the training set of MNIST, then (similarly) a sequence of CIFAR-10 training examples; our model ends up in a weight state W_{MNIST, CIFAR-10} at the end of the entire sequence. No prediction has been done so far. Now using the resulting weights W_{MNIST, CIFAR-10}, we make predictions on examples (called “meta-test test” examples) from the test set of MNIST and CIFAR-10 to evaluate the model’s capability to predict both the first task, MNIST, and the second one, CIFAR-10."
>
> Could you please explain what exactly the reviewer finds "confusing" here?
> We detailed every and each step of the process precisely because the reviewer did not seem to be familiar with few-shot/meta-learning.
> We can also describe standard continual learning with a hand-crafted learning algorithm (let's say, Adam) exactly in the same style:
>
> """
>
> Let’s say our goal is to learn two tasks sequentially: MNIST, then CIFAR-10 using the Adam optimizer.
> We'll train a network with weights W, initialized as W_0.
> We sample input/label example pairs from the training set of MNIST.
> We do a few iterations of Adam on these examples, starting from weight W_0; after this, the weight becomes W_{MNIST}.
> Now we sample input/label example pairs from the training set of CIFAR-10.
> We do a few iterations of Adam on these examples, starting from weight W_{MNIST}; after this, the weight becomes W_{MNIST, CIFAR-10}.
> This is the end of training; the resulting weight is W_{MNIST, CIFAR-10}.
> Now we make predictions on examples from the test set of MNIST and CIFAR-10 to evaluate the model’s (with weight W_{MNIST, CIFAR-10}) capability to predict both the first task, MNIST, and the second one, CIFAR-10.
>
> """
>
> Is this confusing?
> The only difference between the conventional CL and ours is that the learning algorithm used for CL is hand-crafted (e.g, Adam) or learned (forward pass of SRWM).
> We emphasize that in general, we are not supposed to go into this level of detail to describe continual learning though.
>
> > The fact that three independent reviewers had very similar concerns about the proposed approach means something in my opinion
>
> This is not true. Reviewer WhrD wrote "The paper is well written and properly structured. The figures are of high quality and help in quickly grasping the main ideas."
> We also believe that the figure is almost self-explanatory of the method but we also understand that this perception may depend on the readers' familiality with meta-learning/sequence-processing.
>
> > I think that it would be beneficial for the paper if the authors largely re-write the paper, at least the problem formulation, the description of the method, and the setup of the experiments, also following the more common terminology and assumptions in the CL and metalearning literature.
>
> The terminology on metalearning has been already pointed out by Reviewer StyD, and we have immediately adopted his/her suggestion as Reviewer StyD was specific about his/her suggestion.
> We'll only be able to re-write the paper if the reviewer can be more specific: what terminology and assumptions in the CL literature is the reviewer referring to?

---

> > ### Author Response · Authors · 2023-11-21
> >
> > Just to give an external reference in case this might also help: we now use the same terminology as e.g., in Beaulieu et al. https://arxiv.org/abs/2002.09571 as suggested by Reviewer StyD (described in their Sec 2., paragraph starting with *"Before continuing, it is helpful to establish terminology for metalearning, as it is complex."*).

---

> > > ### Author Response · Authors · 2023-11-22
> > >
> > > We'd like to draw the reviewer's attention to the fact that (1) Reviewer WhrD, who has positively rated our writing from the beginning, increased the score, (2) Reviewer StyD, who had suggested a terminology improvement, confirmed that his/her concern on that point has been resolved.
> > >
> > > To make our thought transparent: at this stage, our impression is that some of the reviewer's confusions about our method may stem from the reviewer's general confusion about few-shot/meta-learning via sequence processing (in our view, the reviewer's question/misunderstanding about "temporally structured inputs" seems to support this hypothesis too). For example, is what's described in the reference above, i.e., Sec 2. in Beaulieu et al. https://arxiv.org/abs/2002.09571 all clear to the reviewer?
> > > If the reviewer agrees with us on this observation, we'd kindly ask the reviewer to adjust her/his confidence score.
> > >
> > > In any case, we'll be happy to respond to any other questions. Thank you very much.

---

> > > > ### Comment · Reviewer_jN2L · 2023-11-22
> > > > **My final remark**
> > > >
> > > > Dear Authors, in your latest remarks it seems that you are doubting my understanding/knowledge of this area. Without disclosing my identity, I have published at top-tier conferences in the area of continual learning. And given that your paper claims to make an important contribution in CL, I have to evaluate it based on how that community defines that problem and evaluates proposed solutions.
> > > >
> > > > By the way, I think the rebuttal process for a conference (as opposed to a journal) is supposed to play a very different role: short clarifications, fixing minor issues, adding missing references, etc -- not trying to rewrite large portions of the paper, asking the reviewers to evaluate the paper again in 3-4 days.
> > > >
> > > > So, despite the long argumentation by the authors in the rebuttal, my opinion remains:
> > > > a) this paper does not solve a practically useful formulation of the CL problem
> > > > b) even if we consider it as a solution to some hypothetical problem, it would not scale to a large number of tasks.

---

> > > > > ### Author Response · Authors · 2023-11-22
> > > > >
> > > > > Dear Reviewer jN2L,
> > > > >
> > > > > First of all, if the reviewer found any of our comments offensive in any way, we deeply apologize, that was not at all our intention.
> > > > >
> > > > > > Without disclosing my identity, I have published at top-tier conferences in the area of continual learning.
> > > > >
> > > > > Here we are discussing solely based on the contents of the text provided as reviews and responses. Naturally, if a statement in the review is wrong, we point it out and try to clarify. If a reviewer's suggestion is unclear, we ask to be specific. We do the same whether ​t​he reviewer is an ​expert in all the areas involved in this work or ​somebody with no publication.
> > > > >
> > > > > ​This will also be our final response to ​t​he reviewer.
> > > > >
> > > > > Once again, thank you very much for reviewing our work.

---

### Official Review · Reviewer_WhrD · 2023-11-01

**Soundness:** 2 fair
**Presentation:** 4 excellent
**Contribution:** 2 fair
**Rating:** 6
**Confidence:** 3

**Summary:**

The paper presents a continual learning method based on self-referential weight matrices. By posing the continual learning problem as a meta-learning task, it is possible to formulate the standard continual learning desiderata (low forgetting, high forward and backward transfers) simply as terms of the meta-learning objective. Authors show that their approach is promising through experiments on MNIST, Omniglot, and Mini-ImageNet.

**Strengths:**

Originality is the main strength of the proposed approach. To my knowledge, the application of SRWM to continual learning is a novel idea. Automating the discovery of continual learning algorithms by including the desired requirements as loss terms of the meta-learner is an exciting approach and it would be great to explore it in a bit more details. The paper is well written and properly structured. The figures are of high quality and help in quickly grasping the main ideas.

**Weaknesses:**

The main weakness of the paper is the experimental evaluation. Despite presenting their approach as a continual learning method, the authors don't use any of the standard benchmarks (e.g. Split MNIST, Split Mini-ImageNet), nor do they compare to any previous work (regularization, replay, or parameter isolation methods). The meta-learning formulation is also a major limitation, as the number of loss terms grows rapidly with the number of tasks and it is not clear whether the method is practical for e.g. 10 tasks (which is still a small number compared to the requirements of real-world lifelong learning). It would also be great to include a figure that illustrates the architecture of your model in more detail.

**Questions:**

How do the data requirements of your method grow with the number of tasks?

How is the training sequence constructed? Do you use a single sequence? If not, doesn't it mean you're effectively performing joint training?

Could you elaborate what do you mean by "certain real-world data may naturally give rise to an ACL-like objective"?

---

> ### Author Response · Authors · 2023-11-21
> **Response to Reviewer WhrD**
>
> We thank the reviewer for valuable time reviewing our work and for very positive comments on the originality and clarity of this work.
>
> > Authors show that their approach is promising through experiments on MNIST, Omniglot, and Mini-ImageNet.
>
> Just to clarify as this is not fully accurate: the set of datasets we used is Omniglot, Mini-ImageNet, FC100 for (meta-)training and MNIST, Fashion-MNIST, and CIFAR-10 for (meta-)testing.
>
> > How do the data requirements of your method grow with the number of tasks?
>
> In terms of data requirements to increase the number of tasks, what we need is essentially to increase the number of classes. In principle, we would only need a handful of examples for each class (as we anyway want to train our models as sample-efficient few shot learners). For example, Omniglot has 1632 classes with only 20 examples each.
>
> > How is the training sequence constructed?
>
> We follow the standard procedure used in few-shot learning with sequence processing networks; that is, given a dataset with $C$ classes, for each sequence, we sample $N$ random but distinct classes out of $C$ ($N < C$). The resulting $N$ classes are re-labelled such that each class is assigned to one out of $N$ distinct random label index which is unique to the sequence.
>
> > Do you use a single sequence? If not, doesn't it mean you're effectively performing joint training?
>
> We use multiple sequences in a batch, but no, it does not result in any form of joint training (please also refer to our response to Reviewer jN2L where we clarify more misunderstandings).  This is because the training sequences are constructed such that class-to-label mapping is unique to each sequence (that is, each sequence is a unique learning problem). While this is actually classic in few-shot learning settings, since other reviewers also find this obscure, we’ll be happy to add a more elaborated description in the final version.
>
> > Could you elaborate what do you mean by "certain real-world data may naturally give rise to an ACL-like objective"?
>
> Yes, what we mean is the following. If we consider a language model trained on a very long training sequence (as is the case now with large models); it is not implausible that such a sequence naturally consists of (let’s say three) text chunks (A, B, A’) where A’ is “related” to A (each letter is a chuck/paragraph of text). In this sequence (processed from left to right), learning to predict part A’ after having observed (A, B) encourages remembering the contents of part A. This is very similar to the (artificial) construction of our ACL loss, but obtained without explicit intention of continual learning objective.
>
> We hope the most crucial concern of the reviewer is resolved through the Split-MNIST results.
> Regarding other limitations, we consider them as interesting research questions to be investigated in follow-up works; we believe the current/updated version of the paper demonstrates that the proposed method is an interesting new approach for continual learning. If the reviewer agrees with this, we’d appreciate it a lot if you can consider increasing the score. Thank you very much.
>
> **PS: Please find our response to other questions in "Common Responses to All Reviewers" above. Thank you.**

---

> > ### Comment · Reviewer_WhrD · 2023-11-21
> > **Response to the authors**
> >
> > Thanks for addressing my concerns and running the Split-MNIST experiment. I have raised my score.

---

> > > ### Author Response · Authors · 2023-11-21
> > > **Thank you for the score update**
> > >
> > > We are glad to read that our response was successful.
> > > Thank you very much for the increased score.

---

### Author Response · Authors · 2023-11-20
**General Response**

*(First of all: our apologies as this response comes toward the end of the discussion period; we did not want to only provide superficial responses, e.g., solely emphasizing the originality; we’ve been working on proper extra experiments to address all of the valid concerns of the reviewers. We believe the reviewers will not be disappointed by these results. In case the reviewer has no time to respond before the end of this period, we’d appreciate it if you could still take a look at our response/update for your final decision/recommendation. Thank you very much.)*

We thank all the reviewers for their valuable comments.
We are glad to read that the reviewers found our method original and exciting. Thank you very much for acknowledging this.
The main concern of the reviewers is the lack of evaluations on certain datasets or comparisons to certain baselines. These are valid points, which we address in this rebuttal.

We have just **updated the manuscript**.
While we will precisely reply to each reviewer in the individual reply, here we provide a summary of the most important updates:

* (the most important update) We added extra experiments on **Split-MNIST** (Table 3; text in Sec 4.3 and Appendix A.6).
We compare our ACL models with **8 standard continual learning baselines** as well as one **meta-continual learning approach** (OML).
In short, our ACL models perform better than any of these approaches.
This should resolve the main concerns of many reviewers.
Please check the corresponding results in **Table 3** and text in **Sec 4.3 and Appendix A.6**.

* Regarding Reviewer jN2L’s concerns about whether our method respects the framework of CL, the short answer is yes: we are not affected by ANY of the problems/concerns raised by the reviewer. We will resolve/explain the details of core misunderstandings in the individual response. We believe these confusions are due to some of our terminologies. Following Reviewer StyD’s suggestion, we updated the text with a more standard "meta-training train/test" and "meta-testing train/test" terminology to clearly distinguish meta-training and meta-testing phases.

* We updated the paragraph on "Prior work" (Sec. 5, Page 8) to refer to more prior works on meta-continual learning (while stressing on how they fundamentally differ from our approach) as suggested by Reviewer StyD. One of the references is the baseline used in Table 3 on Split-MNIST.
There was actually also one reference mistake: we had originally claimed that "in-context catastrophic forgetting" has been already pointed out in Munkhdalai and Yu ICML 2017 "Meta networks"; this was actually not correct; they use a standard learning algorithm for continual learning of the meta-learner. We corrected this. In the end, to the best of our knowledge, the study of ‘in-context catastrophic forgetting’ is also novel.

Please note that all important changes are **highlighted in color/blue in the updated PDF** to facilitate the reviewers’ effort to identity what’s new.

We will post detailed/individual replies to each reviewer within the next 24 hours (as well as our code in the supplemental material, requested by Reviewer StyD).
Thank you very much.

---

> ### Author Response · Authors · 2023-11-21
> **Common Responses to All Reviewers**
>
> ### **All Reviewers**
> > (Reviewer WhrD) The main weakness of the paper is the experimental evaluation. Despite presenting their approach as a continual learning method, the authors don't use any of the standard benchmarks (e.g. Split MNIST, Split Mini-ImageNet), nor do they compare to any previous work (regularization, replay, or parameter isolation methods).
>
> > (Reviewer jN2L) I see that the method is significantly different from other continual learning methods, still I would expect the authors to benchmark against some existing methods.
>
> > (Reviewer StyD) There are several important prior works in this domain that were not mentioned in the paper. They should also be compared as baselines.
>
> We agree with the reviewers. To address this very important point, we conducted extra experiments on Split-MNIST (mentioned by both Reviewer WhrD and jN2L). The results can be found in the updated PDF (Table 3 and text in **Sec 4.3 and Appendix A.6). We follow the standard task definition: domain-incremental and class-incremental learning settings (using Hsu et al. 2018 [1]’s terminology).
> This allows us to compare our method to many existing methods, including 8 classic continual learning baselines: standard optimizers SGD & Adam, L2 regularization, two elastic weight consolidation variants, synaptic Intelligence, memory-aware synapses, and learning-without-forgetting (thanks to Hsu et al. 2018 [1]’s overview), as well as one meta-continual learning baseline (Javeh and White 2019 [2]) suggested by Reviewer StyD.
>
> The overall result we obtain is very positive: our out-of-the-box ACL model performs very competitively with the best existing methods (see our note below) in the domain-incremental setting, while it largely outperforms them in the 2-task class-incremental setting. By further fine-tuning this model with the 5-task ACL objective, we obtain a model that largely outperforms all other methods in all the settings. We hope this convinces the reviewers regarding the promise of our method.
>
> Note: Please note that our comparison does not include replay-memory or parameter-increasing methods because such methods are orthogonal to ours; they can be combined to our method to potentially further improve the performance.
>
> [1] Hsu et al., 2018. ​​Re-evaluating continual learning scenarios: A categorization and case for strong baselines. https://arxiv.org/abs/1810.12488
>
> [2] Javeh and White, 2019. Meta-Learning Representations for Continual Learning. https://arxiv.org/abs/1905.12588
>
>
>
> ### **Reviewer WhrD and Reviewer jN2L**
>
> > (Reviewer WhrD) The meta-learning formulation is also a major limitation, as the number of loss terms grows rapidly with the number of tasks and it is not clear whether the method is practical for e.g. 10 tasks (which is still a small number compared to the requirements of real-world lifelong learning).
>
> > (Reviewer jN2L) Another major concern is whether ACL can be used in a practical context in which many tasks will be learned over time (as opposed to just a handful).
> > If you examine Equation 5 on the last page, you'll notice that in order to learn a third task, they need to add three terms to the loss function. In continual learning, a five-task setting is considered small. To learn SplitMNIST, for example, they would actually need 1 + 2 + 3 + 4 + 5 (15) terms in the loss function. As a result, their method becomes quadratically more expensive in terms of computation (i.e., for Task n, you require backpropagation through (n)(n-1)/2 terms). This is clearly not practical.
>
> We first would like to note that computing these loss terms isn’t immediately impractical because they essentially just require forwarding the network for one step, for many independent inputs/images. This can be heavily parallelized as a batch operation.
>
> While this is a valid concern when scaling up more, a natural open research question is: will we really need all these terms in the case we have many more tasks. Our experiments (both the new Table 3 results on Split-MNIST, and the old results on 4 tasks, now Table 4) effectively show that using more terms in the ACL loss improves performance. That said, ideally, we want these models to ‘systematically generalize’ to more tasks even when they are trained with only a handful of them. Indeed, our out-of-the-box model trained only with the 2-task ACL objective does not completely break even when evaluated on the 5-task setting of Split-MNIST (see Table 3; row “ACL (Out-of-the-box model”). We consider this as an interesting research question on generalization to be studied in a follow-up work.

---

### Meta-Review · Area_Chair_D3MW · 2023-12-06

**Metareview:**

This submission addresses the well established Continual Learning (CL) problem, devising a new meta-learning method which “learns how to learn” continually in-context through self-referential weight matrices (which produce self-modifications as auxiliary outputs). In such settings, rather than explicitly designing algorithms to fulfill certain criteria deemed important, the meta-learning method finds an appropriate trade-off between the various competing CL objectives.

Reviewers seem generally positive about the idea of representing continual learning as a sequence learning problem and allowing the outer optimization problem to determine the requirements implicitly, remarking on it being “a fresh perspective” and “an exciting approach”. On the other hand, reviewers seem less optimistic about experimental evaluation, remarking on a lack of comparison to previous work and standard benchmarks (WhrD, StyD), which the authors partially addressed in their response, including “Meta-Learning Representations for Continual Learning. NeurIPS (2019)” but not other methods (“We concluded that comparing to […] is unnecessary here”). While other suggestions e.g. about missing details in task construction were addressed and code was provided, other important suggestions regarding e.g. task sequence length were unfortunately not acted upon. A general promise to “run a 10-task setting for the final version” is appreciated but does not constitute actionable result that can influence the acceptance decision.

Finally, in terms of the categorization of the work, unclarity remains (to me, the argued difference between ACL and MAML based methods is unclear and unconvincing), which is unhelpful in a field as loaded with terminology as Meta/Continual Learning.

In terms of an acceptance decision, we feel that despite the authors’ concern, all reviewers have engaged in an active discussion and provided authors with several opportunities to make a clear case for acceptance (with one reviewer raising their score). In the end, the remaining disagreements and borderline ratings with no clear champion for acceptance among the reviewers let me to conclude that this submission has unfortunately not crossed the threshold for acceptance this time.

**Justification For Why Not Higher Score:**

See Meta Review

**Justification For Why Not Lower Score:**

N/A

---

### Decision · Program_Chairs · 2024-01-16

Reject